# Relaxing the Additivity Constraints in Decentralized No-Regret High-Dimensional Bayesian Optimization

**Anthony Bardou**[*] **& Patrick Thiran**
IC, EPFL
Lausanne, Switzerland
{anthony.bardou,patrick.thiran}@epfl.ch

**Thomas Begin**
ENS Lyon, UCBL, CNRS, LIP
Lyon, France
thomas.begin@ens-lyon.fr

## Abstract

Bayesian Optimization (BO) is typically used to optimize an unknown function $f$ that is noisy and costly to evaluate, by exploiting an acquisition function that must be maximized at each optimization step. Even if provably asymptotically optimal BO algorithms are efficient at optimizing low-dimensional functions, scaling them to high-dimensional spaces remains an open problem, often tackled by assuming an additive structure for $f$. By doing so, BO algorithms typically introduce additional restrictive assumptions on the additive structure that reduce their applicability domain. This paper contains two main contributions: (i) we relax the restrictive assumptions on the additive structure of $f$ *without* weakening the maximization guarantees of the acquisition function, and (ii) we address the over-exploration problem for decentralized BO algorithms. To these ends, we propose DuMBO, an asymptotically optimal decentralized BO algorithm that achieves very competitive performance against state-of-the-art BO algorithms, especially when the additive structure of $f$ comprises high-dimensional factors.

## 1 Introduction

Many real-world applications involve optimizing an unknown, noisy, costly-to-evaluate objective function $f$. Examples of such tasks include hyper parameters tuning in deep neural networks (Bergstra et al., 2013), robotics (Lizotte et al., 2007), networking (Hornby et al., 2006) and computational biology (González et al., 2014). In such applications, $f$ can be seen as a black box that can only be discovered by successive queries. This prevents the use of traditional first-order approaches to optimize $f$.

Bayesian Optimization (BO) has become a highly effective framework for black-box optimization. In general, a BO algorithm tackles this problem by modeling $f$ as a Gaussian process (GP) and by leveraging this model to query $f$ at specific inputs. The challenge of querying $f$ is to trade off exploration (*i.e.* to adequately query an input that improves the quality of the GP regression of $f$) for exploitation (*i.e.* to query an input that is thought to be the maximal argument of $f$). To achieve this trade-off at time $t$, a BO algorithm maximizes an acquisition function $\varphi_t(\boldsymbol{x})$, built by leveraging the information provided by the GP model, to select a query $\boldsymbol{x}^t$.

Although BO has shown its efficiency at optimizing black-box functions, so far it has mostly found success with low-dimensional input spaces (Wang et al., 2013). However, real-world applications, such as computer vision, robotics or networking, often involve a high-dimensional objective function $f$. Scaling BO algorithms to such input spaces remains a great challenge as the cost of finding $\arg\max \varphi_t$ grows exponentially with the input space dimension $d$. A classical way to circumvent that issue is to cap the complexity of the maximization by assuming an additive decomposition of $f$ (e.g. see Kandasamy et al. (2015)) with a low *Maximum Factor Size* (MFS), denoted by $\bar{d}$, which is the maximum number of dimensions for a factor of the decomposition. Unfortunately, assuming an additive decomposition with low MFS may lead to the optimization of a coarse approximation of $f$.

---

[*]Some of this work was done while AB was at ENS Lyon.

The performance of BO algorithms under the assumption of low MFS has been extensively studied (e.g. see Kandasamy et al. (2015); Hoang et al. (2018); Mutny & Krause (2018)), with Hoang et al. (2018) detailing efforts to relax the assumption on MFS. In this paper, we demonstrate that it is possible to completely relax the low-MFS assumptions that limit the applicability domain of asymptotically optimal BO algorithms while still providing provable global maximization guarantees on the acquisition function. To illustrate this, we propose DuMBO, a decentralized, message-passing, asymptotically optimal BO algorithm able to infer a complex additive decomposition of $f$ without any assumption regarding its MFS. As far as we know, this is the first BO algorithm to display such desirable property. Additionally, we provide an efficient way to approximate the well-known GP-UCB acquisition function (Srinivas et al., 2012) in a decentralized context. Finally, we evaluate DuMBO and establish its superiority against several state-of-the-art solutions on both synthetic and real-world problems wherein the noisy objective function $f$ may or may not be decomposed.

## 2 BACKGROUND

### 2.1 STATE OF THE ART

Given a black-box objective function $f : \mathcal{D} \subset \mathbb{R}^d \rightarrow \mathbb{R}$, the goal of a BO algorithm is to find $\boldsymbol{x}^* = \arg\max_{\boldsymbol{x}} f(\boldsymbol{x})$ using as few queries as possible. The quality of the optimization is measured with the immediate regret $r_t = f(\boldsymbol{x}^*) - f(\boldsymbol{x}^t)$, and the cumulative regret $R_t = \sum_{i=1}^{t} r_i$. A BO algorithm is said to be asymptotically optimal if $\lim_{t\rightarrow+\infty} R_t/t = 0$, which implies that the BO algorithm will asymptotically reach $\boldsymbol{x}^*$ and hence guarantees *no-regret* performance.

A BO algorithm typically uses a GP to infer a posterior distribution for the value of $f(\boldsymbol{x})$ at any point $\boldsymbol{x} \in \mathcal{D}$ and selects, at each time step $t$, a query $\boldsymbol{x}^t$. The BO algorithm bases its querying policy on the maximization of an acquisition function that quantifies the benefits of observing $f(\boldsymbol{x})$ in terms of exploration and exploitation. Common acquisition functions include probability of improvement (Jones et al., 1998), expected improvement (Mockus, 1994) and upper confidence bound (Auer, 2003). Like many other acquisition functions, the latter leads to an asymptotically optimal application to GPs, called GP-UCB (Srinivas et al., 2012) and defined as

$$\varphi_t(\boldsymbol{x}) = \mu_t(\boldsymbol{x}) + \beta_t^{\frac{1}{2}} \sigma_t(\boldsymbol{x}). \tag{1}$$

It involves an exploitation term $\mu_t(\boldsymbol{x})$, which is the posterior mean of the GP at input $\boldsymbol{x}$, and an exploration term $\sigma_t(\boldsymbol{x})$, which is the posterior standard deviation of the GP at input $\boldsymbol{x}$. The scalar $\beta_t^{1/2}$ handles the exploration-exploitation trade-off in order to guarantee the asymptotic optimality of GP-UCB with high probability.

As stated before, scaling BO algorithms to high-dimensional functions is challenging because of the exponential complexity of the optimization algorithms used to maximize $\varphi_t$. To tackle this problem, BO algorithms generally fall into one of the two following categories (with the exception of TuRBO proposed by Eriksson et al. (2019), which uses trust regions to maximize $f$).

**Embedding** BO algorithms assume that only a few dimensions significantly impact $f$ and project the high-dimensional space of $f$ into a low-dimensional one where the optimization is actually performed. REMBO (Wang et al., 2016) and ALEBO (Letham et al., 2020) use random matrices to embed the high-dimensional space while SAASBO (Eriksson & Jankowiak, 2021) uses sparse GPs defined on subspaces and LineBO (Kirschner et al., 2019) exploits successive line-searches in random directions. Other approaches such as Gómez-Bombarelli et al. (2018); Moriconi et al. (2020) are based, respectively, on Variational Auto-Encoders and on manifold GPs to learn an embedding. Gupta et al. (2020) propose to perform the optimization in two orthogonal subspaces. Finally, some approaches select a subset of dimensions of the input space to project onto. Such recent methods include Dropout (Li et al., 2017) and MCTS-VS (Song et al., 2022).

**Decomposing** BO algorithms assume an additive structure for $f$ and optimize the factors of the induced decomposition. Classical approaches such as MES (Wang & Jegelka, 2017), ADD-GPUCB (Kandasamy et al., 2015) or QFF (Mutny & Krause, 2018) assume a decomposition with a MFS equal to 1 and orthogonal domains. More recent approaches like DEC-HBO (Hoang et al., 2018) are able to optimize decompositions with larger MFS and shared input components. Still, the MFS of the decomposition must be low to avoid a prohibitive computational complexity. Note that,

Table 1: Comparison of state-of-the-art decomposing BO algorithms with DuMBO on relevant criteria. Here, $n$ is the number of factors in the decomposition, $d$ the number of dimensions of $f$, $\bar{d}$ the MFS of the decomposition, $t$ the optimization step, $\zeta$ the desired accuracy when maximizing $\varphi_t$ and $N_A$ a constant defined in Appendix E. $N_m$ is a constant defined in Hoang et al. (2018).

| Solution | Complexity | MFS Assumption | Find $\arg\max \varphi_t$ |
|---|---|---|---|
| ADD-GPUCB | $\mathcal{O}\left(t^3 + nt^2 + n^2\zeta^{-1}\right)$ | $\bar{d} = 1$ | Yes |
| QFF | $\mathcal{O}\left((\zeta^{-1}t^{3/2}(\log t)^{\bar{d}})^{\bar{d}}\right)$ | $\bar{d} = 1$ | Yes |
| DEC-HBO | $\mathcal{O}\left(N_m\zeta^{-\bar{d}}n(t^3 + n)\right)$ | Low $\bar{d}$ | Under assumptions |
| DuMBO | $\mathcal{O}\left(\bar{d}N_A nt^3\zeta^{-1}\right)$ | None | Yes |

under some assumptions on $f$, these approaches are provably asymptotically optimal and a subset of them, namely ADD-GPUCB (Kandasamy et al., 2015) and DEC-HBO (Hoang et al., 2018), can be used in a decentralized context. Finally, note that in a recent work, (Ziomek & Ammar, 2023) showed that, in an adversarial context, exploiting random decompositions is optimal on average.

### 2.2 DuMBO (Decentralized Message-passing Bayesian Optimization algorithm)

We propose DuMBO, a decomposing algorithm that relaxes the low MFS constraint on the assumed additive decomposition of $f$. Table 1 gathers the main differences between DuMBO and state-of-the-art decomposing algorithms. Note that ADD-GPUCB and QFF require the simplest form of additive decompositions (*i.e.*, $\bar{d} = 1$). As a consequence, when optimizing a complex objective function $f$, they often need to coarsely approximate it. In return, they are able, at each time step $t$, to query $\arg\max \varphi_t$. In contrast, DEC-HBO tolerates more complex decompositions (*i.e.*, with $\bar{d} > 1$), but is no longer guaranteed to find the global maximum of $\varphi_t$, because it uses a max-sum algorithm (Rogers et al., 2011) that requires $f$ to have a sparse additive decomposition to converge. Finally, DuMBO is the only algorithm that is able to completely relax any assumption on the MFS without weakening the maximization guarantees on the acquisition function. This allows DuMBO to be simultaneously asymptotically optimal and able to handle arbitrarily complex decompositions.

The remaining of this article is devoted to formulating the BO problem (Section 3), presenting DuMBO (Section 4), providing theoretical guarantees (Section 5) and comparing its empirical performance with state-of-the-art BO algorithms (Section 6).

## 3 Problem Formulation and First Results

In this section, we introduce the core assumptions about the black-box objective function $f : \mathcal{D} \to \mathbb{R}$ to obtain an additive decomposition (Section 3.1). Next, we exploit these assumptions to derive inference formulas (Section 3.2) and to adapt GP-UCB to a decentralized context (Section 3.3).

### 3.1 Core Assumptions

In order to optimize $f$ in a decentralized fashion, we make several assumptions.

**Assumption 3.1.** *The unknown objective function $f$ can be decomposed into a sum of factor functions $\left(f^{(i)}\right)_{i\in[\![1,n]\!]}$, with compact domains $\left(D^{(i)}\right)_{i\in[\![1,n]\!]}$, such that $\mathcal{D} = \cup_{i=1}^n D^{(i)}$ and*

$$f = \sum_{i=1}^n f^{(i)}. \tag{2}$$

Any decomposition can be represented by a factor graph where each factor and variable node denote, respectively, one of the $n$ factors of the decomposition and one of the $d$ input components of $f$. An edge exists between a factor node $i$ and a variable node $j$ if and only if $f^{(i)}$ uses $x_j$ as an input component. We use $\mathcal{V}_i, i \in [\![1,n]\!]$, and $\mathcal{F}_j, j \in [\![1,d]\!]$, to denote respectively the set of variable

nodes connected to factor node $i$ and the set of factor nodes connected to variable node $j$. Please refer to Appendix A for a detailed example regarding additive decompositions and factor graphs.

To make predictions about the factor functions without any prior knowledge, we need a model that maps the previously collected inputs with their noisy outputs. Denoting $\boldsymbol{x}_{\mathcal{V}_i} = (x_j)_{j \in \mathcal{V}_i}$, let us introduce the following assumption.

**Assumption 3.2.** *Factor functions $f^{(i)}$ are independent $\mathcal{GP}\left(\mu_0^{(i)}, k^{(i)}\left(\boldsymbol{x}_{\mathcal{V}_i}, \boldsymbol{x}'_{\mathcal{V}_i}\right)\right)$, with prior mean $\mu_0^{(i)} = 0$ and covariance function $k^{(i)}$.*

Since $f$ is a sum of independent GPs, Assumption 3.2 implies that $f$ is also $\mathcal{GP}\left(\mu_0, k(\boldsymbol{x}, \boldsymbol{x}')\right)$ with prior mean $\mu_0 = 0$ and covariance function $k(\boldsymbol{x}, \boldsymbol{x}') = \sum_{i=1}^n k^{(i)}\left(\boldsymbol{x}_{\mathcal{V}_i}, \boldsymbol{x}'_{\mathcal{V}_i}\right)$.

Finally, to ensure the no-regret property of DuMBO (see Section 5), we introduce the following assumption on each $k^{(i)}$.

**Assumption 3.3.** *For any $i \in [\![1, n]\!]$, $k^{(i)}$ is an L-Lipschitz, twice differentiable function on $\mathcal{D}^{(i)}$. Furthermore, it exists $H > 0$ such that, for any $\boldsymbol{x}, \boldsymbol{x}' \in \mathcal{D}^{(i)}$ we have*

$$||\nabla^2 k^{(i)}(\boldsymbol{x}, \boldsymbol{x}')||_2 \leq H. \tag{3}$$

Note that Assumption 3.3 is mild: a large class of covariance functions satisfy it, such as the Matérn class (with $\nu \geq 5/2$), the squared-exponential function or even the rational quadratic function. Please refer to Williams & Rasmussen (2006) for details on these covariance functions.

### 3.2 INFERENCE FORMULAS

For any $\boldsymbol{x} \in \mathcal{D}$ and given the previous input queries $(\boldsymbol{x}^1, \cdots, \boldsymbol{x}^t)$, the vector $(f(\boldsymbol{x}), f(\boldsymbol{x}^1), \cdots, f(\boldsymbol{x}^t))^\top$ is Gaussian. Given the $t$-dimensional vector of noisy outputs $\boldsymbol{y} = (y_1, \cdots, y_t)^\top$, with $y_i = f(\boldsymbol{x}^i) + \epsilon$ and $\epsilon$ a centered Gaussian variable of variance $\sigma^2$, the posterior distribution of the factor $f^{(i)}(\boldsymbol{x})$ is also Gaussian. Since $f$ can be decomposed, the posterior mean $\mu_{t+1}^{(i)}(\boldsymbol{x}_{\mathcal{V}_i})$ and variance $(\sigma_{t+1}^{(i)}(\boldsymbol{x}_{\mathcal{V}_i}))^2$ of the factor $f^{(i)}$ at time $t+1$ can be expressed with the posterior means and covariance functions of the factor functions involved in decomposition (2).

**Proposition 3.4.** *Let $\mu_{t+1}^{(i)}(\boldsymbol{x}_{\mathcal{V}_i})$ and $(\sigma_{t+1}^{(i)}(\boldsymbol{x}_{\mathcal{V}_i}))^2$ be the posterior mean and variance of $f^{(i)}$ at input $\boldsymbol{x}_{\mathcal{V}_i}$, respectively. Then, for the decomposition (2),*

$$\mu_{t+1}^{(i)}(\boldsymbol{x}_{\mathcal{V}_i}) = \boldsymbol{k}_{\boldsymbol{x}_{\mathcal{V}_i}}^{(i)\top}\left(\boldsymbol{K} + \sigma^2\boldsymbol{I}\right)^{-1}\boldsymbol{y} \tag{4}$$

$$(\sigma_{t+1}^{(i)}(\boldsymbol{x}_{\mathcal{V}_i}))^2 = k^{(i)}(\boldsymbol{x}_{\mathcal{V}_i}, \boldsymbol{x}_{\mathcal{V}_i}) - \boldsymbol{k}_{\boldsymbol{x}_{\mathcal{V}_i}}^{(i)\top}\left(\boldsymbol{K} + \sigma^2\boldsymbol{I}\right)^{-1}\boldsymbol{k}_{\boldsymbol{x}_{\mathcal{V}_i}}^{(i)} \tag{5}$$

*with $t \times 1$ vectors $\boldsymbol{k}_{\boldsymbol{x}_{\mathcal{V}_i}}^{(i)} = (k^{(i)}(\boldsymbol{x}_{\mathcal{V}_i}, \boldsymbol{x}_{\mathcal{V}_i}^j))_{j \in [\![1,t]\!]}$, $t \times t$ matrix $\boldsymbol{K} = (k(\boldsymbol{x}^j, \boldsymbol{x}^k))_{j,k \in [\![1,t]\!]}$ and $\boldsymbol{I}$ the $t \times t$ identity matrix.*

For the sake of generality, Proposition 3.4 only requires an additive decomposition of $f$. Appendix B describes how such a decomposition can be inferred from data, using the method proposed by Gardner et al. (2017). Note that Proposition 3.4 does *not* assume a corresponding additive decomposition of the observed outputs in $\boldsymbol{y}$. However a large class of real-world applications naturally come up with such an additive output decomposition (e.g., network throughput maximization (Bardou & Begin, 2022), energy consumption minimization (Bourdeau et al., 2019) or UAVs-related applications (Xie et al., 2018)). As shown by Wang et al. (2020), having access to a decomposed output can only improve the predictive performance of the GP surrogate model. Therefore, we derive the inference formulas to handle the case where the output decomposition is known in Appendix C. Also, we explore the benefits of having access to the decomposed output of $f$ in Section 6.

### 3.3 PROPOSED ACQUISITION FUNCTION

Having defined a surrogate model for $f$, we now turn to finding an optimal policy for querying the objective function. In this section, we exploit the decomposition of $f$ and its associated factor graph (see Appendix A) to build an acquisition function for our BO algorithm that approximates GP-UCB in a decentralized context. Proofs for all the presented results can be found in Appendix D.

Recall that GP-UCB is defined by (1) as the sum of an exploitation term $\mu_t(\boldsymbol{x})$ and an exploration term $\sigma_t(\boldsymbol{x})$ weighted by some scalar $\beta_t^{1/2}$. Finding an additive decomposition for GP-UCB is hard, because Assumption 3.1 allows $\mu_t(\boldsymbol{x})$ to be expressed as a sum, but not $\sigma_t(\boldsymbol{x})$. To circumvent this caveat, Kandasamy et al. (2015) proposed to apply GP-UCB to each factor of the additive decomposition of $f$, with $\varphi_t^{(i)} = \mu_t^{(i)} + \beta_t^{1/2}\sigma_t^{(i)}$. Then, they proved that their algorithm ADD-GPUCB offers no-regret performance by taking $\sum_{i=1}^n \varphi_t^{(i)} = \mu_t + \beta_t^{1/2}\sum_{i=1}^n \sigma_t^{(i)}$ as the acquisition function. Although the exploitation term $\mu_t$ is preserved, the exploration term is now overweighted since $\sum_{i=1}^n \sigma_t^{(i)} \geq \sqrt{\sum_{i=1}^n \left(\sigma_t^{(i)}\right)^2} = \sigma_t$. To reach better empirical performance, one could look for a tighter additive upper bound of $\sigma_t^2$. This is the purpose of this section.

In a given factor graph, a factor node $i$ can access information about another factor node if they share a common variable node $j$ (see Figure 3 in Appendix A). We gather all the indices of the factor nodes that share at least one variable node with the factor node $i$ in $\mathcal{N}_i = \cup_{j\in\mathcal{V}_i}\mathcal{F}_j$. Then, we propose the following approximation for $\sigma_t(\boldsymbol{x})$:

$$\sum_{i=1}^n \sqrt{\sum_{k\in\mathcal{N}_i} \frac{\left(\sigma_t^{(k)}(\boldsymbol{x}_{\mathcal{V}_k})\right)^2}{|\mathcal{N}_k|^2}}. \tag{6}$$

On the one hand, this approximation is exact and equal to $\sigma_t(\boldsymbol{x})$ for a complete factor graph (i.e., $\forall i \in [\![1,n]\!], |\mathcal{N}_i| = n$). On the other hand, given a decomposition made only of one-dimensional factors with orthogonal domains (i.e., $\forall i \in [\![1,n]\!], |\mathcal{N}_i| = 1$), it boils down to the approximation proposed by Kandasamy et al. (2015), that is, $\sum_{i=1}^n \sigma_t^{(i)}(\boldsymbol{x}_{\mathcal{V}_i})$. A benefit of approximation (6) is to better exploit the structure of the factor graph. Indeed, the following result shows that (6) is a tighter upper bound of $\sigma_t(\boldsymbol{x})$ than the one proposed in Kandasamy et al. (2015).

**Theorem 3.5.** *Let Assumptions 3.1 and 3.2 hold. Then, for any factor graph and any $\boldsymbol{x} \in \mathcal{D}$,*

$$\sigma_t(\boldsymbol{x}) \leq \sum_{i=1}^n \sqrt{\sum_{k\in\mathcal{N}_i} \frac{\left(\sigma_t^{(k)}(\boldsymbol{x}_{\mathcal{V}_k})\right)^2}{|\mathcal{N}_k|^2}} \leq \sum_{i=1}^n \sigma_t^{(i)}(\boldsymbol{x}_{\mathcal{V}_i}). \tag{7}$$

From the bounds of Theorem 3.5, one can expect the acquisition function

$$\varphi_t(\boldsymbol{x}) = \mu_t(\boldsymbol{x}) + \beta_t^{\frac{1}{2}}\sum_{i=1}^n \sqrt{\sum_{k\in\mathcal{N}_i} \frac{\left(\sigma_t^{(k)}(\boldsymbol{x}_{\mathcal{V}_k})\right)^2}{|\mathcal{N}_k|^2}} \tag{8}$$

to be less prone to over-exploration. Thus, a BO algorithm using (8) behaves more like GP-UCB than ADD-GPUCB or DEC-HBO. Note that (8) has a natural decomposition $\sum_{i=1}^n \varphi_t^{(i)}(\boldsymbol{x}_{\mathcal{V}_i})$ with

$$\varphi_t^{(i)}(\boldsymbol{x}_{\mathcal{V}_i}) = \mu_t^{(i)}(\boldsymbol{x}_{\mathcal{V}_i}) + \beta_t^{\frac{1}{2}}\sqrt{\frac{\left(\sigma_t^{(i)}(\boldsymbol{x}_{\mathcal{V}_i})\right)^2}{|\mathcal{N}_i|^2} + c_i} \tag{9}$$

with $c_i = \sum_{\substack{k\in\mathcal{N}_i \\ k\neq i}} \frac{\left(\sigma_t^{(k)}(\boldsymbol{x}_{\mathcal{V}_k})\right)^2}{|\mathcal{N}_k|^2}$ computed by message-passing with the variable nodes in $\mathcal{V}_i$.

## 4 DUMBO

In this section, we describe DuMBO, a BO algorithm that exploits the results from Section 3 to find $\arg\max_{\boldsymbol{x}\in\mathcal{D}} \sum_{i=1}^n \varphi_t^{(i)}(\boldsymbol{x}_{\mathcal{V}_i})$. Optimizing $\varphi_t(\boldsymbol{x}) = \sum_{i=1}^n \varphi_t^{(i)}(\boldsymbol{x}_{\mathcal{V}_i})$ while ensuring the consistency between shared input components is equivalent to solving the constrained optimization problem

$$\max \sum_{i=1}^n \varphi_t^{(i)}\left(\boldsymbol{x}^{(i)}\right) \text{ such that } \boldsymbol{x}_{\mathcal{V}_i\cap\mathcal{V}_j}^{(i)} = \boldsymbol{x}_{\mathcal{V}_i\cap\mathcal{V}_j}^{(j)}, \forall i,j \in [\![1,n]\!] \tag{10}$$

with $\boldsymbol{x}^{(1)}, \cdots, \boldsymbol{x}^{(n)}$ being the inputs (whose dimension indices are respectively listed in $\mathcal{V}_1, \cdots, \mathcal{V}_n$) of the factor functions $\varphi_t^{(1)}, \cdots, \varphi_t^{(n)}$.

To simplify the equality constraints in (10), we introduce a consensus variable $\bar{\boldsymbol{x}} \in \mathcal{D}$ and we reformulate the optimization problem as

$$\max \sum_{i=1}^{n} \varphi_t^{(i)} \left( \boldsymbol{x}^{(i)} \right) \text{ such that } \boldsymbol{x}^{(i)} = \bar{\boldsymbol{x}}_{\mathcal{V}_i}, \forall i \in [\![1, n]\!]. \tag{11}$$

We now turn the problem (11) into an unconstrained optimization problem by considering its augmented Lagrangian $\mathcal{L}_\eta(\boldsymbol{x}^{(1)}, \cdots, \boldsymbol{x}^{(n)}, \bar{\boldsymbol{x}}, \boldsymbol{\lambda})$:

$$\mathcal{L}_\eta = \sum_{i=1}^{n} \varphi_t^{(i)}(\boldsymbol{x}^{(i)}) - \boldsymbol{\lambda}^{(i)\top}(\boldsymbol{x}^{(i)} - \bar{\boldsymbol{x}}_{\mathcal{V}_i}) - \frac{\eta}{2}||\boldsymbol{x}^{(i)} - \bar{\boldsymbol{x}}_{\mathcal{V}_i}||_2^2 \tag{12}$$

with $\boldsymbol{\lambda}_k^{(i)}$ a column vector of dual variables with $|\mathcal{V}_i|$ components and a hyperparameter $\eta > 0$, which can be set dynamically following the procedure detailed in Boyd et al. (2011).

To maximize (12), we consider the Alternating Direction Method of Multipliers (ADMM), proposed by Gabay & Mercier (1976). ADMM is an iterative method that proposes, at iteration $k$, to solve sequentially the problems

$$\boldsymbol{x}_{k+1}^{(1)} = \underset{\boldsymbol{x}^{(1)}}{\arg\max} \, \mathcal{L}(\boldsymbol{x}^{(1)}, \cdots, \boldsymbol{x}_k^{(n)}, \bar{\boldsymbol{x}}_k, \boldsymbol{\lambda}_k)$$

$$\vdots$$

$$\boldsymbol{x}_{k+1}^{(n)} = \underset{\boldsymbol{x}^{(n)}}{\arg\max} \, \mathcal{L}(\boldsymbol{x}_{k+1}^{(1)}, \cdots, \boldsymbol{x}_{k+1}^{(n-1)}, \boldsymbol{x}^{(n)}, \bar{\boldsymbol{x}}_k, \boldsymbol{\lambda}_k)$$

$$\bar{\boldsymbol{x}}_{k+1} = \underset{\bar{\boldsymbol{x}}}{\arg\max} \, \mathcal{L}(\boldsymbol{x}_{k+1}^{(1)}, \cdots, \boldsymbol{x}_{k+1}^{(n)}, \bar{\boldsymbol{x}}, \boldsymbol{\lambda}_k) \tag{13}$$

$$\boldsymbol{\lambda}_{k+1} = \underset{\boldsymbol{\lambda}}{\arg\max} \, \mathcal{L}(\boldsymbol{x}_{k+1}^{(1)}, \cdots, \boldsymbol{x}_{k+1}^{(n)}, \bar{\boldsymbol{x}}_{k+1}, \boldsymbol{\lambda}). \tag{14}$$

Note that $\boldsymbol{x}_{k+1}^{(1)}, \cdots, \boldsymbol{x}_{k+1}^{(n)}$ can be found concurrently by each factor node of the factor graph of $f$. We propose to proceed by gradient ascent (e.g. with ADAM (Kingma & Ba, 2015)) of

$$\mathcal{L}_\eta^{(i)} = \varphi_t^{(i)}(\boldsymbol{x}^{(i)}) - \boldsymbol{\lambda}^{(i)\top}(\boldsymbol{x}^{(i)} - \bar{\boldsymbol{x}}_{\mathcal{V}_i}) - \frac{\eta}{2}||\boldsymbol{x}^{(i)} - \bar{\boldsymbol{x}}_{\mathcal{V}_i}||_2^2. \tag{15}$$

Next, each factor node $i$ sends $\left( \boldsymbol{x}_{k+1}^{(i)}, \frac{\left( \sigma_t^{(i)}(\boldsymbol{x}_{k+1}^{(i)}) \right)^2}{|\mathcal{N}_i|^2} \right)$ to its variable nodes in $\mathcal{V}_i$. Each variable node $j$ uses the received data to compute (13) and (14). In fact, if $\forall i \in [\![1, n]\!]$, $\sum_{j \in \mathcal{F}_i} \lambda_{0,i}^{(j)} = 0$, it is known (see Boyd et al. (2011)) that the closed-forms for (13) and (14) are

$$\bar{\boldsymbol{x}}_{k+1} = \left( \frac{1}{|\mathcal{F}_i|} \sum_{j \in \mathcal{F}_i} x_{k+1,i}^{(j)} \right)_{i \in [\![1,d]\!]} \tag{16}$$

$$\boldsymbol{\lambda}_{k+1} = \left( \boldsymbol{\lambda}_k^{(i)} + \eta \left( \boldsymbol{x}_{k+1}^{(i)} - \bar{\boldsymbol{x}}_{k+1,\mathcal{V}_i} \right) \right)_{i \in [\![1,n]\!]}. \tag{17}$$

Finally, each variable node $j$ sends $\left( \boldsymbol{\lambda}_{k+1}^{(i)}, \bar{x}_{k+1,j} \right)$ as well as $\left( \frac{\left( \sigma_t^{(l)}(\boldsymbol{x}_{k+1}^{(l)}) \right)^2}{|\mathcal{N}_l|^2} \right)_{l \in \mathcal{F}_j}$ to its factor node $i$, $i \in \mathcal{F}_j$. This allows each factor node $i$ to update its dual variables $\boldsymbol{\lambda}^{(i)}$ as well as the value of term $c_i$ in (9).

These results describe a fully decentralized message-passing algorithm, called DuMBO, which can run on the factor graph of $f$. The detailed algorithm (Algorithm 1), as well as a discussion about its time complexity, are provided in Appendix E. Since DuMBO relies on ADMM to maximize $\varphi_t$, let us briefly discuss its maximization guarantees. It is well known that ADMM converges towards the global maximum of a convex $\varphi_t$. ADMM has also demonstrated very good performance at optimizing non-convex functions (Liavas & Sidiropoulos, 2015; Lai & Osher, 2014; Chartrand & Wohlberg, 2013). This is explained by recent works such as Wang et al. (2019), which extends the global maximization guarantee of ADMM to the class of *restricted prox-regular* functions that satisfy the Kurdyka-Lojasiewicz condition. We demonstrate that the acquisition function $\varphi_t$ simultaneously satisfies these conditions in the next section.

## 5 ASYMPTOTIC OPTIMALITY

In this section, we demonstrate the asymptotic optimality of DuMBO. First, we prove that, at each iteration, ADMM is always able to globally maximize the acquisition function $\varphi_t$ with Theorem 5.1. Then, we demonstrate that DuMBO has a lower immediate regret than another asymptotically optimal BO algorithm with Theorem 5.2. With these two theorems, we establish the asymptotical optimality of DuMBO, stated in Corollary 5.3.

Let us start with the global maximization guarantee of ADMM, whose proof can be found in Appendix F.

**Theorem 5.1.** *Under Assumption 3.3, $\forall i \in [\![1, n]\!], \varphi_t^{(i)}$ (see (9)) is a restricted-prox regular function. Furthermore, the augmented Lagrangian $\mathcal{L}_\eta$ (see (12)) is a Kurdyka-Lojasiewicz (KL) function.*

Now that we have the guarantee that $\varphi_t$ is always maximized, we can properly establish the asymptotic optimality of DuMBO. We start by providing an upper bound on its immediate regret $r_t = f(\boldsymbol{x}^*) - f(\boldsymbol{x}_t)$ for a finite, discrete domain $\mathcal{D}$. Its proof can be found in Appendix G.

**Theorem 5.2.** *Let $r_t = f(\boldsymbol{x}^*) - f(\boldsymbol{x}^t)$ denote the immediate regret of DuMBO. Let $\delta \in (0, 1)$ and $\beta_t = 2 \log \left( \frac{|\mathcal{D}|\pi^2 t^2}{6\delta} \right)$. Then $\forall t \in \mathbb{N}$, with probability at least $1 - \delta$,*

$$ r_t \leq 2\beta_t^{\frac{1}{2}} \sum_{i=1}^{n} \sqrt{\sum_{k \in \mathcal{N}_i} \frac{\left( \sigma_t^{(k)}(\boldsymbol{x}_{\mathcal{V}_k}) \right)^2}{|\mathcal{N}_k|^2}}, \tag{18} $$

We demonstrate the asymptotic optimality of DuMBO by piggybacking on the asymptotic optimality of DEC-HBO (Hoang et al., 2018). The latter is a decomposing BO algorithm with an immediate regret bound of $2\beta_t^{1/2} \sum_{i=1}^{n} \sigma_t^{(i)}(\boldsymbol{x}^t)$ over a finite, discrete domain (see Theorem 1 in Hoang et al. (2018)). Interestingly, Theorem 3.5 directly implies that the immediate regret bound (18) is lower than the immediate regret bound of DEC-HBO. As a consequence, the immediate regret of DuMBO is bounded from above by the regret bound of DEC-HBO. This allows us to rely on proofs in Hoang et al. (2018) to establish some properties of DuMBO. In particular, DEC-HBO is provably asymptotically optimal whether the domain $\mathcal{D}$ is discrete or continuous (see Theorems 2 and 3 in Hoang et al. (2018)). These results rely on their immediate regret bound over a finite, discrete domain. Hence, they directly apply to DuMBO as well and yield the following corollary.

**Corollary 5.3.** *Let $\delta \in (0, 1)$ and $R_t = \sum_{k=1}^{t} r_k$ denote the cumulative regret of DuMBO. Then, with probability at least $1 - \delta$, there exists a monotonically increasing sequence $\{\beta_t\}_t$ such that $\beta_t \in \mathcal{O}(\log t)$ and $\lim_{t \to +\infty} R_t/t = 0$.*

## 6 PERFORMANCE EXPERIMENTS

In this section, we detail the experiments carried out to evaluate the empirical performance of DuMBO. An open-source implementation of DuMBO, based on BoTorch (Balandat et al., 2020), is available on GitHub[1].

---

[1] https://github.com/abardou/dumbo

Table 2: Comparison of eight state-of-the-art solutions against two different versions of DuMBO on synthetic and real-world problems. Decomposing BO algorithms can be identified with the prefix "(+)". The reported metrics are the minimal regret attained for the synthetic functions, and the average negative reward for the real-world problems. The significantly best performance metrics among all the strategies are written in **bold text**, and the significantly best among the strategies that do not have access to the additive decomposition are underlined.

| Algorithm | Synthetic Functions $(d\text{-}\bar{d})$ | | | | Real-World Problems $(d\text{-}\bar{d})$ | | |
|---|---|---|---|---|---|---|---|
| *Unknown Add. Dec.* | SHC (2-2) | Hartmann (6-6) | Powell (24-4) | Rastrigin (100-5) | Cosmo (9-) | WLAN (12-6) | Rover (60-) |
| SAASBO | 0.013 | 0.89 | 3,901 | 1,073 | 16.55 | -116.40 | 10.82 |
| TuRBO | 0.322 | 1.89 | 667 | 1,109 | **5.82** | -118.39 | **7.01** |
| LineBO | 0.016 | 0.69 | 4,830 | 1,388 | **5.90** | -118.68 | 8.24 |
| MS-UCB | 0.012 | 0.80 | 22,271 | 1,455 | **5.87** | -117.95 | 7.65 |
| (+) ADD-GPUCB | 0.102 | 1.29 | 11,760 | N/A | 7.46 | -119.05 | 26.57 |
| (+) DEC-HBO | **0.005** | 1.47 | 7,937 | N/A | 14.90 | -116.58 | 10.07 |
| (+) DuMBO | **0.006** | **0.54** | 496 | 986 | **5.86** | -120.67 | **6.38** |
| *Known Add. Dec.* | | | | | | | |
| (+) ADD-DuMBO | 0.009 | **0.53** | **469** | **678** | N/A | **-121.11** | N/A |

Our benchmark comprises four synthetic functions and three real-world experiments. We consider two state-of-the-art decomposing BO algorithms: ADD-GPUCB (Kandasamy et al., 2015) that assumes $\bar{d} = 1$, and DEC-HBO (Hoang et al., 2018) for which, similarly to its authors in their empirical evaluation, we assume $\bar{d} \leq 3$. We also consider four state-of-the-art BO algorithms that do not assume an additive decomposition of the objective function: TuRBO (Eriksson et al., 2019), SAASBO (Eriksson & Jankowiak, 2021), LineBO (Kirschner et al., 2019) and MS-UCB (Gupta et al., 2020) with its hyperparameter $\alpha = 0$. We compare these eight algorithms with two versions of the proposed algorithm: DuMBO that must systematically infer an additive decomposition of $f$ (see Appendix B) and ADD-DuMBO that, conversely, can observe the true decomposition of $f$ if it exists (see Appendix C). Finally, note that we chose a Matérn kernel (with its hyperparameter $\nu = 5/2$) for each GP involved in these experiments.

Since BO is often used in the optimization of expensive black-box functions, we are interested in the ability of each algorithm at obtaining good performance in a small number of iterations. Also, to strengthen our results, each experiment is replicated 5 independent times. Table 2 gathers the averaged results that were obtained. Additionally, we made wall-clock time measurements on some experiments and we discuss them in Appendix J.

## 6.1 OPTIMIZING SYNTHETIC FUNCTIONS

In this section, we compare the six BO algorithms mentioned above using four synthetic functions: the 2d Six-Hump Camel (SHC), the 6d Hartmann, the 24d Powell and the 100d Rastrigin. A detailed description of the synthetic functions, as well as the complete set of figures depicting the performance of the BO algorithms can be found in Appendix H.

Figure 1(a) reports the minimal regrets of the algorithms on the Powell function, where $d = 24$ and the MFS $\bar{d} = 4$. Observe that the two decomposing algorithms, ADD-GPUCB and DEC-HBO, obtain the worst minimal regrets. This is because they infer an additive decomposition of $f$ based on an assumption on the MFS, that is $\bar{d} \leq 3$ when actually $\bar{d} = 4$. Conversely, DuMBO, which does not make any restrictive assumption on $\bar{d}$, manages to rapidly achieve a low regret by inferring an efficient additive decomposition of $f$. DuMBO also outperforms SAASBO, TuRBO, LineBO and MS-UCB. Finally, Figure 1(a) shows that, when given access to the true additive decomposition of $f$, ADD-DuMBO achieves its lowest regret in a lower number of iterations. Similar results were obtained with the optimization of other synthetic functions (see Appendix H). Finally, note that among all the BO algorithms tested in the experiments, the two versions of DuMBO are the only ones able to properly infer and/or exploit the additive decomposition of $f$ given its large MFS.

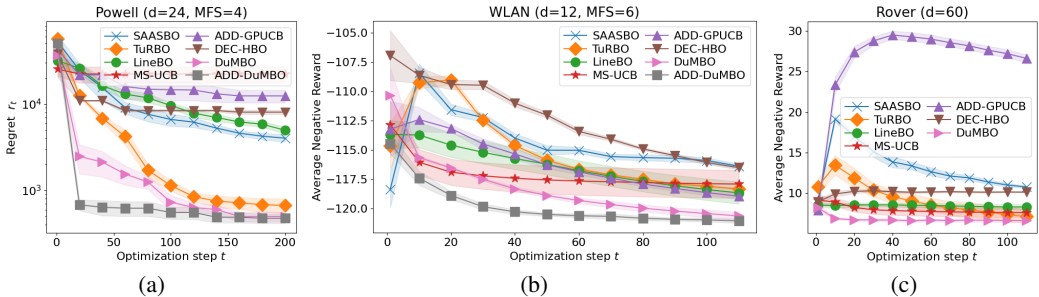

Figure 1: Performance achieved by the BO algorithms listed in Section 6 for (a) the 24d Powell synthetic function, (b) the optimization of the Shannon capacity in a WLAN and (c) the trajectory planning of a rover. The shaded areas indicate the standard error intervals.

## 6.2 SOLVING REAL-WORLD PROBLEMS

We consider three real-world problems: (a) fine-tuning some cosmological constants to maximize the likelihood of observed astronomical data (Cosmo), (b) controlling the power of devices in a Wireless Local Area Network (WLAN) to maximize its Shannon capacity (Kemperman, 1974) and (c) the trajectory planning of a rover (Rover). The problems, along with a complete set of figures depicting the performance of the tested BO algorithms, are discussed in details in Appendix I.

Figures 1(b) and 1(c) depict the performance of the BO algorithms on problems (b) and (c), where $d = 12$ and 60, respectively. Figure 1(b) shows that DuMBO is able to significantly outperform every other state-of-the-art BO algorithm. Additionally, and similarly to what was observed with Figure 1(a), Figure 1(b) suggests that having access, and being able to handle additive decompositions with large MFS, is a significant advantage. As a matter of fact, this allows to outperform BO algorithms that are unable to exploit this additional information. Figure 1(c) exhibits patterns similar to Figure 1(a): ADD-GPUCB and DEC-HBO fail to infer an adequate additive decomposition because of their restrictive MFS assumptions. In contrast, DuMBO, which does not make such an assumption on the size of the MFS, achieves the best performance along with TuRBO. Note that ADD-DuMBO is not evaluated on problem (c) since its objective function is not additive.

## 7 CONCLUSION

In this article, we showed that it is possible to completely relax the restrictive assumptions of low-MFS in the additive decomposition of $f$ without weakening the asymptotic optimality guarantees of decomposing BO algorithms. This allows BO algorithms to simultaneously keep their no-regret property and infer a complex additive decomposition of the objective function $f$, or directly exploit it when it is available. To illustrate the effectiveness of such design choices, we proposed DuMBO, an asymptotically optimal decentralized BO algorithm that optimizes $f$ using a tighter decentralized approximation of GP-UCB that requires less exploration than the previously proposed approximations. As demonstrated by Sections 5 and 6, DuMBO is a no-regret, competitive alternative to state-of-the-art BO algorithms, able to optimize complex objective functions in a small number of iterations. Compared to other decomposing algorithms, such as ADD-GPUCB and DEC-HBO, DuMBO brings a significant improvement, particularly when the decomposition of $f$ has a large MFS with numerous factors.

For future work, we plan to extend DuMBO to batch mode (Li et al., 2016; Daxberger & Low, 2017) and to apply it to suitable technological contexts such as computer networks (Bardou & Begin, 2022), UAVs (Xie et al., 2018) or within a robots team (Chen et al., 2013).

### ACKNOWLEDGMENTS

This work was supported in part by the LABEX MILYON (ANR-10-LABX-0070) of Université de Lyon, within the program "Investissements d'Avenir" (ANR-11-IDEX- 0007) operated by the French National Research Agency (ANR).

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

# A FACTOR GRAPH

In this appendix, we provide an example of an additive decomposition and its associated factor graph. Consider the following additive decomposition

$$f(\boldsymbol{x}) = f^{(1)}(x_1, x_3) + f^{(2)}(x_2) + f^{(3)}(x_2, x_3) + f^{(4)}(x_1, x_3). \qquad (19)$$

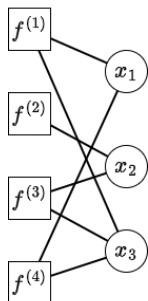

Figure 2: The factor graph of the decomposition (19). The factor nodes and variable nodes are depicted with squares and circles, respectively. In this decomposition, there are $n = 4$ factors and $d = 3$ variables.

The associated factor graph is shown in Figure 2. Observe for instance that, since the first factor $f_1$ exploits the first and third components of the input $\boldsymbol{x}$, the first factor node $f_1$ is connected to the first and third variable nodes $x_1$ and $x_3$.

Using this graph, it is very easy to build $\mathcal{V}_i$ ($1 \leq i \leq n$), the set of variable indices associated with a factor $i$. In fact, $\mathcal{V}_i$ comprises the indices of the variable nodes connected to the factor node $i$. As an example, $\mathcal{V}_1 = \{1, 3\}$. Similarly, it is trivial to build $\mathcal{F}_j$ ($1 \leq j \leq d$), the set of factor indices associated with a variable $j$. In fact, $\mathcal{F}_j$ comprises the indices of the factor nodes connected to the variable node $j$. As an example, $\mathcal{F}_3 = \{1, 3, 4\}$ since factors $f_1$, $f_3$ and $f_4$ exploit $x_3$.

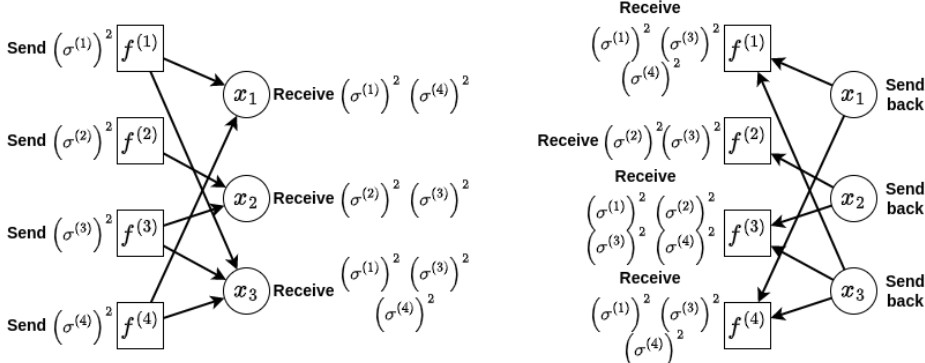

Figure 3: Message passing in the factor graph. (Left) The factor nodes compute their variance terms and send them to their variable nodes. Thus, each variable node $j$ receives the variance terms of the factor nodes in $\mathcal{F}_j$. (Right) The variable nodes send all the collected variance terms to each of their factor nodes. Thus, each factor node $i$ receives the variance terms of the factor nodes in $\mathcal{N}_i = \cup_{j \in \mathcal{V}_i} \mathcal{F}_j$.

In Section 3.3, we introduce a new set for a factor node $i$, denoted by $\mathcal{N}_i$, which is the set of indices of the factors that share at least one component of $\boldsymbol{x}$ with the $i$th factor. $\mathcal{N}_i$ can be easily built from the quantities $\{\mathcal{V}_i\}_{i \in [\![1,n]\!]}$ and $\{\mathcal{F}_j\}_{j \in [\![1,d]\!]}$ as $\mathcal{N}_i = \cup_{j \in \mathcal{V}_i} \mathcal{F}_j$. Note that, by construction, $i \in \mathcal{N}_i$. Using the factor graph in Figure 2 as an example, we can see that $\mathcal{N}_3 = \{2, 3, 4\}$. In practice, the sets $\{\mathcal{N}_i\}_{i \in [\![1,n]\!]}$ can be computed with message-passing, as illustrated by Figure 3.

## B    INFERENCE OF THE ADDITIVE DECOMPOSITION

Our decentralized algorithm requires an additive decomposition of the objective function $f$, as specified in Assumption 3.1 and exploited in Proposition 3.4. If the decomposition is known, it can be directly specified to DuMBO. If the decomposition is unknown, it can be inferred from the data (Gardner et al., 2017; Wang et al., 2017; Lu et al., 2022; Durrande et al., 2012). Also note that in a recent work, Ziomek & Ammar (2023) showed that, in an adversarial context, using random decompositions is optimal on average. Since adversarial contexts are not the primary focus of this paper, we consider the approach introduced by (Gardner et al., 2017), and we briefly discuss it in the following.

As in Hoang et al. (2018), let us associate each candidate additive decomposition $\mathcal{A}$ with the kernel of an additive GP (Duvenaud et al., 2011). Given $k$ candidates $\mathcal{A}_1, \cdots, \mathcal{A}_k$, we reformulate the acquisition function $\varphi_t$ as a weighted average with respect to the posterior of each candidate given the dataset $\mathcal{S} = \{(\boldsymbol{x}_i, y_i)\}_{i \in [\![1,t]\!]}$ composed of the selected input queries and their observed noisy outputs, that is

$$\varphi_t(\boldsymbol{x}) = \sum_{i=1}^{k} p(\mathcal{A}_i | \mathcal{S}) \varphi_t^{\mathcal{A}_i}(\boldsymbol{x}) \tag{20}$$

$$= \sum_{i=1}^{k} p(\mathcal{A}_i | \mathcal{S}) \sum_{j=1}^{|\mathcal{A}_i|} \varphi_t^{(j)}(\boldsymbol{x}_{\mathcal{V}_j}) \tag{21}$$

$$\approx \frac{1}{k} \sum_{i=1}^{k} \sum_{j=1}^{|\mathcal{A}_i|} \varphi_t^{(j)}(\boldsymbol{x}_{\mathcal{V}_j}), \tag{22}$$

with $\varphi_t^{\mathcal{A}_i}$ our proposed acquisition function given the additive decomposition $\mathcal{A}_i$ and $\varphi_t^{(j)}$ given by (9). Moreover, note that (21) follows from (20) since the additive decomposition $\mathcal{A}_i$ also provides an additive decomposition of our proposed acquisition function, and (22) follows from (21) as proposed by Gardner et al. (2017).

The candidates $\mathcal{A}_1, \cdots, \mathcal{A}_k$ are sampled by Monte-Carlo Markov Chain (MCMC) with the Metropolis-Hastings algorithm (Robert et al., 2010), starting from the fully dependent decomposition $\mathcal{A}_0 = \{\{1, \cdots, d\}\}$ at $t = 0$. When the decomposition is unknown, at each time step $t$, $k$ promising decompositions are sampled by MCMC starting from the last sampled decomposition at time step $t - 1$, and (22) is maximized by our decentralized algorithm to find a promising input $\boldsymbol{x}$ to query.

## C    INFERENCE FORMULAS WITH DECOMPOSED OUTPUT

The decentralized algorithm DuMBO requires an additive decomposition of the objective function $f$, but does not assume that the corresponding decomposition of the output is observable. Nevertheless, as demonstrated by Wang et al. (2020), having access to such a decomposed output improves the regression capabilities of the surrogate GP model, mostly by reducing the variance of its predictions. In this appendix, we derive the counterparts of the posterior mean (4) and the posterior variance (5) when the output decomposition of $f$ is observable.

Observing the output decomposition of $f$ means that $f$ is now a function $\mathbb{R}^d \to \mathbb{R}^n$, where $n$ is the number of factors in its additive decomposition. At time $t$, a BO algorithm has access to a $t \times n$ matrix $\boldsymbol{Y}$ instead of a $t$-dimensional output vector $\boldsymbol{y}$, such that $\boldsymbol{Y}\boldsymbol{1} = \boldsymbol{y}$, where $\boldsymbol{1}$ is the $n$-dimensional all-1 vector.

Having access to the matrix $\boldsymbol{Y}$ allows to train $n$ different GPs instead of a single one with an additive kernel, so that the $i$th $\mathcal{GP}\left(0, k^{(i)}\left(\boldsymbol{x}_{\mathcal{V}_i}, \boldsymbol{x}'_{\mathcal{V}_i}\right)\right)$ serves as a surrogate model only for the $i$th factor of the decomposition of $f$. To condition the $i$th GP, we consider the data set $\mathcal{S}_i = \left\{\left(\boldsymbol{x}^j_{\mathcal{V}_i}, Y_{j,i}\right)\right\}_{j \in [\![1,t]\!]}$.

Given $\mathcal{S}_i$, the expressions of the posterior mean $\mu^{(i)}_{t+1}$ and the posterior variance $\left(\sigma^{(i)}_{t+1}\right)^2$ are simple

instances of the conditioned Gaussian distribution formulas, where

$$\mu_{t+1}^{(i)}(\boldsymbol{x}) = \boldsymbol{k}_{\boldsymbol{x}_{\mathcal{V}_i}}^{(i)\top} \left(\boldsymbol{K}_{(i)} + \sigma_i^2 \boldsymbol{I}\right)^{-1} \boldsymbol{Y}_{:i}, \tag{23}$$

$$\left(\sigma_{t+1}^{(i)}(\boldsymbol{x})\right)^2 = k^{(i)}\left(\boldsymbol{x}_{\mathcal{V}_i}, \boldsymbol{x}_{\mathcal{V}_i}\right) - \boldsymbol{k}_{\boldsymbol{x}_{\mathcal{V}_i}}^{(i)\top} \left(\boldsymbol{K}_{(i)} + \sigma_i^2 \boldsymbol{I}\right)^{-1} \boldsymbol{k}_{\boldsymbol{x}_{\mathcal{V}_i}}^{(i)}, \tag{24}$$

with $\boldsymbol{Y}_{:i}$ the $i$th column of $\boldsymbol{Y}$, $t \times 1$ vectors $\boldsymbol{k}_{\boldsymbol{x}_{\mathcal{V}_i}}^{(i)} = (k^{(i)}(\boldsymbol{x}_{\mathcal{V}_i}, \boldsymbol{x}_{\mathcal{V}_i}^j))_{j \in [\![1,t]\!]}$, $t \times t$ matrices $\boldsymbol{K}_{(i)} = (k^{(i)}(\boldsymbol{x}_{\mathcal{V}_i}^j, \boldsymbol{x}_{\mathcal{V}_i}^k))_{j,k \in [\![1,t]\!]}$ and $\boldsymbol{I}$ the $t \times t$ identity matrix.

Equations (23) and (24) differ from (4) and (5) mainly by their ability to exploit the inverse of the Gram matrix built only from the $i$th covariance function $k^{(i)}$, and of course, the outputs of the $i$th factor of the decomposition $\boldsymbol{Y}_{:i}$.

## D  PROPOSED ACQUISITION FUNCTION

In this appendix, we prove Theorem 3.5. Let us start by the following lemma.

**Lemma D.1.** *For all factor graphs and for all* $\boldsymbol{x} \in \mathcal{D}$,

$$\sum_{i=1}^{n} \sqrt{\sum_{k \in \mathcal{N}_i} \frac{\left(\sigma_t^{(k)}(\boldsymbol{x}_{\mathcal{V}_k})\right)^2}{|\mathcal{N}_k|^2}} \leq \sum_{i=1}^{n} \sigma_t^{(i)}(\boldsymbol{x}_{\mathcal{V}_i}). \tag{25}$$

*Proof.*

$$\sum_{i=1}^{n} \sqrt{\sum_{k \in \mathcal{N}_i} \frac{\left(\sigma_t^{(k)}(\boldsymbol{x}_{\mathcal{V}_k})\right)^2}{|\mathcal{N}_k|^2}} \leq \sum_{i=1}^{n} \sum_{k \in \mathcal{N}_i} \frac{\sigma_t^{(k)}(\boldsymbol{x}_{\mathcal{V}_k})}{|\mathcal{N}_k|} \tag{26}$$

$$= \sum_{i=1}^{n} \sigma_t^{(i)}(\boldsymbol{x}_{\mathcal{V}_i}) \sum_{k \in \mathcal{N}_i} \frac{1}{|\mathcal{N}_i|} \tag{27}$$

$$= \sum_{i=1}^{n} \sigma_t^{(i)}(\boldsymbol{x}_{\mathcal{V}_i}). $$

Equation (26) comes from a simple application of the triangle inequality with the euclidean norm. Equation (27) follows by construction of each $\mathcal{N}_k$, as the term $\frac{\sigma_t^{(k)}(\boldsymbol{x}_{\mathcal{V}_k})}{|\mathcal{N}_k|}$ appears exactly $|\mathcal{N}_k|$ times within the double sum of the RHS (right hand side) of (26). This concludes the proof.  □

Now, let us prove the second lemma that leads to Theorem 3.5.

**Lemma D.2.** *For all factor graphs and for all* $\boldsymbol{x} \in \mathcal{D}$,

$$\sigma_t(\boldsymbol{x}) \leq \sum_{i=1}^{n} \sqrt{\sum_{k \in \mathcal{N}_i} \frac{\left(\sigma_t^{(k)}(\boldsymbol{x}_{\mathcal{V}_k})\right)^2}{|\mathcal{N}_k|^2}}. \tag{28}$$

*Proof.* First, let us denote the LHS (left hand side) of (28) by $L(\sigma_t^{(1)}, \cdots, \sigma_t^{(n)})$ and the RHS of (28) by $R(\sigma_t^{(1)}, \cdots, \sigma_t^{(n)})$. Note that the arguments of the variance terms have been omitted for the sake of simplicity. Next, let us differentiate both $L$ and $R$ with respect to each variance term $\left(\sigma_t^{(k)}\right)^2$. We find

$$\frac{\partial L}{\partial \left(\sigma_t^{(k)}\right)^2} = \frac{1}{2\sqrt{\sum_{i=1}^{n}\left(\sigma_t^{(i)}\right)^2}}, \tag{29}$$

$$\frac{\partial R}{\partial \left(\sigma_t^{(k)}\right)^2} = \sum_{i \in \mathcal{N}_k} \frac{1}{2|\mathcal{N}_k|^2 \sqrt{\sum_{j \in \mathcal{N}_i} \frac{\left(\sigma_t^{(j)}\right)^2}{|\mathcal{N}_j|^2}}}. \tag{30}$$

Let us now compare (29) and (30) by studying their ratio, denoted $Q_k$,

$$Q_k = \left( \sum_{i \in \mathcal{N}_k} \frac{\sqrt{\sum_{j=1}^n \left(\sigma_t^{(j)}\right)^2}}{|\mathcal{N}_k|^2 \sqrt{\sum_{j \in \mathcal{N}_i} \frac{\left(\sigma_t^{(j)}\right)^2}{|\mathcal{N}_j|^2}}} \right)^{-1}$$

$$\leq \left( \sum_{i=1}^n \frac{\sqrt{\sum_{j=1}^n \left(\sigma_t^{(j)}\right)^2}}{n^2 \sqrt{\sum_{j=1}^n \frac{\left(\sigma_t^{(j)}\right)^2}{n^2}}} \right)^{-1} \tag{31}$$

$$= \left( \frac{1}{n} \sum_{i=1}^n \frac{\sqrt{\sum_{j=1}^n \left(\sigma_t^{(j)}\right)^2}}{\sqrt{\sum_{j=1}^n \left(\sigma_t^{(j)}\right)^2}} \right)^{-1}$$

$$= 1 \tag{32}$$

where (31) comes from the fact that $Q_k$ is maximized when $|\mathcal{N}_i| = n, \forall i \in [1, n]$.

Since $\forall k \in [1, n], Q_k \leq 1$, we have $0 \leq \frac{\partial L}{\partial \left(\sigma_t^{(k)}\right)^2} \leq \frac{\partial R}{\partial \left(\sigma_t^{(k)}\right)^2}$. This inequality can be exploited by observing that

$$L(\sigma_t^{(1)}, \cdots, \sigma_t^{(n)}) = L(0, \cdots, 0) + \sum_{i=1}^n \int_0^{\left(\sigma_t^{(i)}\right)^2} \frac{\partial L}{\partial \left(\sigma_t^{(i)}\right)^2} d\left(\sigma_t^{(i)}\right)^2$$

$$= \sum_{i=1}^n \int_0^{\left(\sigma_t^{(i)}\right)^2} \frac{\partial L}{\partial \left(\sigma_t^{(i)}\right)^2} d\left(\sigma_t^{(i)}\right)^2 \tag{33}$$

$$\leq \sum_{i=1}^n \int_0^{\left(\sigma_t^{(i)}\right)^2} \frac{\partial R}{\partial \left(\sigma_t^{(i)}\right)^2} d\left(\sigma_t^{(i)}\right)^2 \tag{34}$$

$$= R(\sigma_t^{(1)}, \cdots, \sigma_t^{(n)}) \tag{35}$$

where (33) and (35) follow from $L(0, \cdots, 0) = R(0, \cdots, 0) = 0$, and (34) comes from the inequality involving the partial derivatives of $L$ and $R$ established above.

This directly implies that for any $\boldsymbol{x} \in \mathcal{D}$ and any factor graph, (7) is greater than (or equal to) $\sigma_t(\boldsymbol{x})$. $\qquad\square$

Combining Lemmas D.1 and D.2 together yields Theorem 3.5.

# E  DuMBO: Algorithm and Time Complexity

In this section, we describe DuMBO in Algorithm 1, alongside its time complexity.

We now provide a time complexity analysis for Algorithm 1. The analysis assumes that a gradient ascent performs $\mathcal{O}\left(\zeta^{-1}\right)$ steps for a desired accuracy $\zeta$ (Xie et al., 2020) and ADMM converges in at most $N_A$ steps. We also denote by $d^{(i)}$ the factor size of the $i$th factor in the decomposition, used by the local acquisition function $\varphi_t^{(i)}$. Note that, within the factor graph of $\varphi_t$, $n$ factor nodes and $d$ variable nodes work concurrently to run ADMM in a decentralized fashion. We provide the time complexities for the two types of nodes in this section (the communication costs between factor nodes and variable nodes are neglected for the clarity of the analysis).

---

**Algorithm 1** DuMBO

---

1: **Input**: factor graph of the acquisition function $\varphi_t$, sequence of $\beta_t$, $\eta > 0$.
2: $t = 0$
3: **while true do**
4:     $\boldsymbol{\lambda}_0 = \mathbf{0}$
5:     $\forall i \in [\![1, n]\!]$, init $\boldsymbol{x}_0^{(i)}$ randomly
6:     $k = 0$
7:     **while** stopping criterion not met **do**
8:       **for all** factor node $i$ [concurrently] **do**
9:         **if** $k \neq 0$ **then**
10:           Update $\boldsymbol{\lambda}_k^{(i)}$ with (17) and the received data
11:           Compute $c_i = \sum_{\substack{l \in \mathcal{N}_i \\ l \neq i}} \frac{\left(\sigma_t^{(l)}(\boldsymbol{x}_k^{(l)})\right)^2}{|\mathcal{N}_l|^2}$ with the received data
12:         **end if**
13:         Compute $\boldsymbol{x}_{k+1}^{(i)}$ by maximizing (15) with gradient ascent starting from $\boldsymbol{x}_k^{(i)}$
14:         Send $x_{k+1,j}^{(i)}$ and $\frac{\left(\sigma_t^{(i)}(\boldsymbol{x}_{k+1}^{(i)})\right)^2}{|\mathcal{N}_i|^2}$ to the variable node $j$, $\forall j \in \mathcal{V}_i$
15:       **end for**
16:       **for all** variable node $j$ [concurrently] **do**
17:         Compute $\bar{x}_{k+1,j}$ with (16)
18:         Send $\bar{x}_{k+1,j}$ and $\left\{ \frac{\left(\sigma_t^{(i)}(\boldsymbol{x}_{k+1}^{(i)})\right)^2}{|\mathcal{N}_i|^2} \right\}_{i \in \mathcal{F}_j}$ to the factor nodes in $\mathcal{F}_j$
19:       **end for**
20:       $k = k + 1$
21:     **end while**
22:     Observe $y^{(t+1)} = f(\bar{\boldsymbol{x}}_k) + \epsilon, \epsilon \sim \mathcal{N}\left(0, \sigma^2\right)$
23:     $t = t + 1$
24: **end while**

---

**Factor node.** For a factor node $i$, it is known that, at iteration $t$, the time complexity of the inference with a GP is $\mathcal{O}\left(t^3 d^{(i)}\right)$, where $t$ denotes the number of previous observations. Thus, the time complexity of evaluating (15) is $\mathcal{O}\left(t^3 d^{(i)}\right)$. Since the evaluation is required $\mathcal{O}\left(\zeta^{-1}\right)$ times by the gradient ascent, the time complexity of finding $x_{k+1}^{(i)}$ is $\mathcal{O}\left(\zeta^{-1} t^3 d_m^{(i)}\right)$. A factor node also needs to compute $\boldsymbol{\lambda}_{k+1}^{(i)}$, which is $\mathcal{O}\left(d^{(i)}\right)$. Since the factor node is called at least once and at most $N_A$ times for ADMM to converge, the time complexity of a factor node is $\mathcal{O}\left(d^{(i)} \zeta^{-1} t^3 N_A\right)$.

**Variable node.** A variable node $j$ is simply in charge of collecting messages from $|\mathcal{F}_j|$ factor nodes, and to aggregate them into $\bar{x}_{k+1,j}$ by averaging. Its time complexity is therefore $\mathcal{O}\left(|\mathcal{F}_j|\right)$.

## F   RESTRICTED PROX-REGULARITY AND KL PROPERTY OF THE ACQUISITION FUNCTION

In this section, we prove Theorem 5.1, which is split in two lemmas (Lemmas F.2 and F.4) on the acquisition function and the augmented Lagrangian, respectively. For the sake of completeness, we provide the definition of the property to prove in each lemma before stating the lemma itself.

Let us recall the definition of restricted prox-regularity.

**Definition F.1** (Restricted Prox-Regularity (Wang et al., 2019))**.** *For a lower semi-continuous function $\varphi_t^{(i)}$, let $M \in \mathbb{R}_+$, and define the exclusion set*

$$S_M = \left\{ \boldsymbol{x} \in \mathcal{D}^{(i)} : ||\boldsymbol{g}||_2 > M, \forall \boldsymbol{g} \in \partial \varphi_t^{(i)}(\boldsymbol{x}) \right\} \tag{36}$$

*with $\partial \varphi_t^{(i)}(\boldsymbol{x})$ the set of all subgradients of $\varphi_t^{(i)}(\boldsymbol{x})$.*

$\varphi_t^{(i)}$ is called restricted prox-regular if, for any $M > 0$ and bounded set $T \subseteq \mathcal{D}^{(i)}$, there exists $\gamma > 0$ such that

$$\varphi_t^{(i)}(\boldsymbol{x}') + \frac{\gamma}{2}||\boldsymbol{x}' - \boldsymbol{x}||_2^2 \geq \varphi_t^{(i)}(\boldsymbol{x}) + \boldsymbol{g}^\top(\boldsymbol{x}' - \boldsymbol{x}), \forall \boldsymbol{x} \in T \setminus S_M, \boldsymbol{x}' \in T, \boldsymbol{g} \in \partial\varphi_t^{(i)}(\boldsymbol{x}). \quad (37)$$

If $T \setminus S_M = \emptyset$, (37) is automatically satisfied.

**Lemma F.2.** *Under Assumption 3.3, for any $i \in [\![1, n]\!]$, $\varphi_t^{(i)}$ (see (9)) is restricted-prox regular.*

*Proof.* First, note that $\varphi_t^{(i)}$, given by (9), is indeed a continuous function, and therefore a lower semi-continuous function as required by Wang et al. (2019).

Then, let us pick $M > 0$, any bounded set $T \subseteq \mathcal{D}^{(i)}$ and build the corresponding exclusion set $S_M$ with (36). Under Assumption 3.3 and for any $\boldsymbol{x} \in T \setminus S_M$, (9) is differentiable. Therefore, for any $\boldsymbol{x} \in \mathcal{D}^{(i)}$, $\partial\varphi_t^{(i)}(\boldsymbol{x}) = \left\{\nabla\varphi_t^{(i)}(\boldsymbol{x})\right\}$. $\nabla\varphi_t^{(i)}(\boldsymbol{x})$ is directly obtained by differentiation of (9), which also involves differentiating (4) and (5).

$$\nabla\varphi_t^{(i)}(\boldsymbol{x}) = \nabla\boldsymbol{k}^{(i)}(\boldsymbol{x}, \boldsymbol{X})^\top \left(\boldsymbol{K} + \sigma^2\boldsymbol{I}\right)^{-1}\boldsymbol{y} - \frac{\beta_t^{1/2}}{|\mathcal{N}_i|}\frac{\nabla\boldsymbol{k}^{(i)}(\boldsymbol{x}, \boldsymbol{X})^\top\left(\boldsymbol{K} + \sigma^2\boldsymbol{I}\right)^{-1}\boldsymbol{k}^{(i)}(\boldsymbol{x}, \boldsymbol{X})}{D(\boldsymbol{x})^{1/2}}$$
$$(38)$$

with $\nabla\boldsymbol{k}^{(i)}(\boldsymbol{x}, \boldsymbol{X})$ being the $t \times d$ matrix $\left(\nabla k^{(i)}(\boldsymbol{x}, \boldsymbol{x}^1), \cdots, \nabla k^{(i)}(\boldsymbol{x}, \boldsymbol{x}^t)\right)^\top$, $\lambda^{(i)} = k^{(i)}(\boldsymbol{x}, \boldsymbol{x})$ and

$$D(\boldsymbol{x}) = \lambda^{(i)} - \boldsymbol{k}^{(i)}(\boldsymbol{x}, \boldsymbol{X})^\top\left(\boldsymbol{K} + \sigma^2\boldsymbol{I}\right)^{-1}\boldsymbol{k}^{(i)}(\boldsymbol{x}, \boldsymbol{X}) + |\mathcal{N}_i|^2 c_i. \quad (39)$$

Observe that, for any $\boldsymbol{x} \in T \setminus S_M$, we have $|D(\boldsymbol{x})| \geq D_M$, for some $D_M > 0$. This is because, otherwise, $||\nabla\varphi_t^{(i)}(\boldsymbol{x})||_2$ could diverge, which is impossible for $\boldsymbol{x} \in T \setminus S_M$. Under Assumption 3.3, (38) is also differentiable. For any $M > 0$, any bounded set $T \subseteq \mathcal{D}^{(i)}$, any $\boldsymbol{x} \in T \setminus S_M$ and any $\boldsymbol{x}' \in T$, we have

$$\varphi_t^{(i)}(\boldsymbol{x}') - \varphi_t^{(i)}(\boldsymbol{x}) - \nabla\varphi_t^{(i)}(\boldsymbol{x})^\top(\boldsymbol{x}' - \boldsymbol{x}) \geq -\frac{||\nabla^2\varphi_t^{(i)}(\boldsymbol{x})||_2}{2}||\boldsymbol{x}' - \boldsymbol{x}||_2^2 \quad (40)$$

with $\nabla^2\varphi_t^{(i)}(\boldsymbol{x})$ the Hessian matrix of $\varphi_t^{(i)}$ at point $\boldsymbol{x}$.

We will now bound $||\nabla^2\varphi_t^{(i)}(\boldsymbol{x})||_2$ from above by decomposing $\nabla^2\varphi_t^{(i)}(\boldsymbol{x}) = \boldsymbol{A}(\boldsymbol{x}) + \frac{\beta_t^{1/2}}{|\mathcal{N}_i|}(\boldsymbol{B}(\boldsymbol{x}) + \boldsymbol{C}(\boldsymbol{x}))$ with

$$\boldsymbol{A}(\boldsymbol{x}) = \nabla^2\boldsymbol{k}^{(i)}(\boldsymbol{x}, \boldsymbol{X})\left(\boldsymbol{K} + \sigma^2\boldsymbol{I}\right)^{-1}\boldsymbol{y}, \quad (41)$$

$$\boldsymbol{B}(\boldsymbol{x}) = \frac{1}{D(\boldsymbol{x})^{1/2}}\nabla^2\boldsymbol{k}^{(i)}(\boldsymbol{x}, \boldsymbol{X})\left(\boldsymbol{K} + \sigma^2\boldsymbol{I}\right)^{-1}\boldsymbol{k}^{(i)}(\boldsymbol{x}, \boldsymbol{X}) + \nabla\boldsymbol{k}^{(i)}(\boldsymbol{x}, \boldsymbol{X})^\top\left(\boldsymbol{K} + \sigma^2\boldsymbol{I}\right)^{-1}\nabla\boldsymbol{k}^{(i)}(\boldsymbol{x}, \boldsymbol{X}),$$
$$(42)$$

$$\boldsymbol{C}(\boldsymbol{x}) = \frac{1}{D(\boldsymbol{x})^{3/2}}\left(\nabla\boldsymbol{k}^{(i)}(\boldsymbol{x}, \boldsymbol{X})^\top\left(\boldsymbol{K} + \sigma^2\boldsymbol{I}\right)^{-1}\boldsymbol{k}^{(i)}(\boldsymbol{x}, \boldsymbol{X})\right)^2, \quad (43)$$

where $\nabla^2\boldsymbol{k}^{(i)}(\boldsymbol{x}, \boldsymbol{X})$ is the $d \times d \times t$ tensor resulting from stacking the Hessian matrices $\nabla^2 k^{(i)}(\boldsymbol{x}, \boldsymbol{x}^j)$ ($j \in [\![1, t]\!]$) on the third axis of the tensor.

We can now bound $||\boldsymbol{A}(\boldsymbol{x})||_2$, $||\boldsymbol{B}(\boldsymbol{x})||_2$ and $||\boldsymbol{C}(\boldsymbol{x})||_2$ from above for any $\boldsymbol{x} \in T \setminus S_M$. In particular, note that under Assumption 3.3, the Pythagorean theorem yields $||\nabla^2 k^{(i)}(\boldsymbol{x})||_2 = \sqrt{t}H$. Also, note

that $||\boldsymbol{k}^{(i)}(\boldsymbol{x}, \boldsymbol{X})||_2 \leq \sqrt{t}\lambda^{(i)}$. Therefore, we have

$$||\boldsymbol{A}(\boldsymbol{x})||_2 \leq \sqrt{t}H|| \left(\boldsymbol{K} + \sigma^2 \boldsymbol{I}\right)^{-1} ||_2 ||\boldsymbol{y}||_2, \tag{44}$$

$$||\boldsymbol{B}(\boldsymbol{x})||_2 \leq \frac{1}{D_M^{1/2}} \left( \sqrt{t}H|| \left(\boldsymbol{K} + \sigma^2 \boldsymbol{I}\right)^{-1} ||_2 \sqrt{t}\lambda^{(i)} + L|| \left(\boldsymbol{K} + \sigma^2 \boldsymbol{I}\right)^{-1} ||_2 L \right)$$

$$= \frac{|| \left(\boldsymbol{K} + \sigma^2 \boldsymbol{I}\right)^{-1} ||_2}{D_M^{1/2}} \left( tH\lambda^{(i)} + L^2 \right), \tag{45}$$

$$||\boldsymbol{C}(\boldsymbol{x})||_2 \leq \frac{L^2|| \left(\boldsymbol{K} + \sigma^2 \boldsymbol{I}\right)^{-1} ||_2^2 t \left(\lambda^{(i)}\right)^2}{D_M^{3/2}} \tag{46}$$

Note that the occurrences of $L$ in (45) and (46) are due to $k^{(i)}$ being an $L$-Lipschitz function (see Assumption 3.3).

Combining (44), (45) and (46), we have an upper bound for $||\nabla^2 \varphi_t^{(i)}(\boldsymbol{x})||_2$ for any $\boldsymbol{x} \in T \setminus S_M$:

$$||\nabla^2 \varphi_t^{(i)}(\boldsymbol{x})||_2 \leq \sqrt{t}|| \left(\boldsymbol{K} + \sigma^2 \boldsymbol{I}\right)^{-1} ||_2 \left( H||\boldsymbol{y}||_2 + \frac{\beta_t^{1/2}\sqrt{t}}{|\mathcal{N}_i| D_M^{1/2}} \left( H\lambda^{(i)} + L^2 \left( 1 + \frac{|| \left(\boldsymbol{K} + \sigma^2 \boldsymbol{I}\right)^{-1} ||_2 \left(\lambda^{(i)}\right)^2}{D_M^2} \right) \right) \right).$$
$$\tag{47}$$

Let us denote by $\gamma$ the RHS of (47). Then, for any $M > 0$, any $T \subseteq \mathcal{D}^{(i)}$, any $\boldsymbol{x} \in T \setminus S_M$ and any $\boldsymbol{x}' \in T$, we have

$$\varphi_t^{(i)}(\boldsymbol{x}') - \varphi_t^{(i)}(\boldsymbol{x}) - \nabla\varphi_t^{(i)}(\boldsymbol{x})^\top (\boldsymbol{x}' - \boldsymbol{x}) \geq -\frac{\gamma}{2}||\boldsymbol{x}' - \boldsymbol{x}||_2^2,$$

which is equivalent to

$$\varphi_t^{(i)}(\boldsymbol{x}') + \frac{\gamma}{2}||\boldsymbol{x}' - \boldsymbol{x}||_2^2 \geq \varphi_t^{(i)}(\boldsymbol{x}) + \nabla\varphi_t^{(i)}(\boldsymbol{x})^\top (\boldsymbol{x}' - \boldsymbol{x}). \tag{48}$$

This concludes the proof. $\qquad\square$

Now let us focus on the last part of Theorem 5.1, regarding the augmented Lagrangian $\mathcal{L}_\eta$ (see (12)). We must prove that $\mathcal{L}_\eta$ is a KL function, so let us first provide the definition for the sake of completeness.

**Definition F.3** (Kurdyka-Łojasiewicz Function (Attouch et al., 2010)). *A function $\mathcal{L}$ satisfies the Kurdyka-Łojasiewicz (KL) inequality in $\boldsymbol{x} \in dom\, \partial\mathcal{L}$ if there exist $\gamma \in (0, +\infty]$, a neighborhood $U$ of $\boldsymbol{x}$ and a continuous concave function $\xi : [0, \gamma) \to \mathbb{R}_+$, such that:*

- *$\xi(0) = 0$,*

- *$\xi$ is $C^1$ on $(0, \gamma)$,*

- *$\forall s \in (0, \gamma), \xi'(s) > 0$,*

- *$\forall \boldsymbol{x}' \in U \cap \{\boldsymbol{x}' | \mathcal{L}(\boldsymbol{x}) < \mathcal{L}(\boldsymbol{x}') < \mathcal{L}(\boldsymbol{x}) + \gamma\}$, the KL inequality holds*

$$\xi' \left(\mathcal{L}(\boldsymbol{x}') - \mathcal{L}(\boldsymbol{x})\right) \cdot dist\left(0, \partial\mathcal{L}(\boldsymbol{x}')\right) \geq 1 \tag{49}$$

*A function satisfying the KL inequality $\forall \boldsymbol{x} \in dom\, \partial\mathcal{L}$ is called a KL function.*

**Lemma F.4.** *$\mathcal{L}_\eta$ (see (12)) is a KL function.*

*Proof.* First of all, observe that $\mathcal{L}_\eta$ is continuous, hence a lower semi-continuous function. Also, note that $\mathcal{L}_\eta$ is either not subdifferentiable (because of the square root in (9)) or differentiable. Therefore, $\mathcal{L}_\eta$ is differentiable on dom $\partial\mathcal{L}_\eta$ and (49) can be rewritten as

$$\xi' \left(\mathcal{L}_\eta(\boldsymbol{x}') - \mathcal{L}_\eta(\boldsymbol{x})\right) ||\nabla\mathcal{L}_\eta(\boldsymbol{x}')||_2 \geq 1, \forall \boldsymbol{x} \in dom\, \partial\mathcal{L}_\eta. \tag{50}$$

(i) Let us first pick a noncritical point of $\mathcal{L}_\eta$. It is known (see Lemma 2 in Attouch et al. (2010)) that for any noncritical point $\boldsymbol{x}$ of $\mathcal{L}_\eta$, there is $c > 0$ such that, for any $\boldsymbol{x}' \in \operatorname{dom} \mathcal{L}_\eta$, we have

$$||\boldsymbol{x}' - \boldsymbol{x}||_2 + |\mathcal{L}_\eta(\boldsymbol{x}') - \mathcal{L}_\eta(\boldsymbol{x})| < c \implies ||\nabla \mathcal{L}_\eta(\boldsymbol{x}')||_2 \geq c. \tag{51}$$

The point $\boldsymbol{x}$ being a noncritical point of $\mathcal{L}_\eta$, (49) is verified with $U = \{\boldsymbol{x}' : \boldsymbol{x}' \in \operatorname{dom} \mathcal{L}_\eta, |\mathcal{L}_\eta(\boldsymbol{x}') - \mathcal{L}_\eta(\boldsymbol{x})| < c - ||\boldsymbol{x}' - \boldsymbol{x}||_2\}$, $\gamma = c$ and $\xi(s) = \frac{2}{\sqrt{c}}\sqrt{s}$. Indeed then $\xi'(s) = \frac{1}{\sqrt{cs}}$, and for any $\boldsymbol{x}' \in U \cap \{\boldsymbol{z} | \mathcal{L}_\eta(\boldsymbol{x}) < \mathcal{L}_\eta(\boldsymbol{z}) < \mathcal{L}_\eta(\boldsymbol{x}) + \gamma\}$, we have

$$\begin{aligned} \xi'\left(\mathcal{L}_\eta(\boldsymbol{x}') - \mathcal{L}_\eta(\boldsymbol{x})\right) \cdot ||\nabla \mathcal{L}_\eta(\boldsymbol{x}')||_2 &= \frac{||\nabla \mathcal{L}_\eta(\boldsymbol{x}')||_2}{\sqrt{c(\mathcal{L}_\eta(\boldsymbol{x}') - \mathcal{L}_\eta(\boldsymbol{x}))}} \\ &\geq \frac{c}{\sqrt{c^2}} \\ &= 1 \end{aligned} \tag{52}$$

(ii) Regarding the critical points of $\mathcal{L}_\eta$, we use another well-known result (see Theorem ŁI in Kurdyka (1998)) stating that, given an open bounded subset $\Omega$ of $\mathbb{R}^n$, for any function $g : \Omega \to \mathbb{R}$ differentiable on $\Omega \setminus g^{-1}(0)$, there exist $c > 0$, $\gamma > 0$ and $0 \leq \alpha < 1$ such that

$$||\nabla g(\boldsymbol{x})||_2 \geq c|g(\boldsymbol{x})|^\alpha. \tag{53}$$

for any $\boldsymbol{x} \in \Omega$ such that $|g(\boldsymbol{x})| \in (0, \gamma)$.

Let $g(\boldsymbol{x}) = \mathcal{L}_\eta(\boldsymbol{x}) - \mathcal{L}_\eta(\boldsymbol{x}^*)$, where $\boldsymbol{x}^*$ is a critical point of $\mathcal{L}_\eta$. Then, $g$ is differentiable on all $\boldsymbol{x} \in \operatorname{dom} \partial\mathcal{L}_\eta$. Therefore, there exist $c > 0$, $\gamma > 0$ and $0 \leq \alpha < 1$ such that, for any $\boldsymbol{x} \in U = \{x' | |\mathcal{L}_\eta(\boldsymbol{x}') - \mathcal{L}_\eta(\boldsymbol{x}^*)| < \gamma\}$,

$$||\nabla \mathcal{L}_\eta(\boldsymbol{x})||_2 = ||\nabla g(\boldsymbol{x})||_2 \geq c|g(\boldsymbol{x})|^\alpha = c|\mathcal{L}_\eta(\boldsymbol{x}) - \mathcal{L}_\eta(\boldsymbol{x}^*)|^\alpha, \tag{54}$$

since $\boldsymbol{x}^*$ is a critical point. This yields

$$\frac{||\nabla \mathcal{L}_\eta(\boldsymbol{x})||_2}{c|\mathcal{L}_\eta(\boldsymbol{x}) - \mathcal{L}_\eta(\boldsymbol{x}^*)|^\alpha} \geq 1 \tag{55}$$

Choosing $\boldsymbol{x} \in U \cap \{\boldsymbol{x}' | \mathcal{L}_\eta(\boldsymbol{x}^*) < \mathcal{L}_\eta(\boldsymbol{x}') < \mathcal{L}_\eta(\boldsymbol{x}^*) + \gamma\}$ and $\xi(s) = \frac{1}{c(1-\alpha)}s^{1-\alpha}$ (so that $\xi'(s) = \frac{1}{cs^\alpha}$) allows us to rewrite (55) as

$$\xi'\left(\mathcal{L}_\eta(\boldsymbol{x}) - \mathcal{L}_\eta(\boldsymbol{x}^*)\right) \cdot ||\nabla \mathcal{L}_\eta(\boldsymbol{x})||_2 = \frac{||\nabla \mathcal{L}_\eta(\boldsymbol{x})||_2}{c\left(\mathcal{L}_\eta(\boldsymbol{x}) - \mathcal{L}_\eta(\boldsymbol{x}^*)\right)^\alpha} \geq 1. \tag{56}$$

Combining (52) and (56) concludes our proof. $\qquad\square$

## G IMMEDIATE REGRET BOUND

In this section, we discuss the asymptotic optimality of DuMBO and we provide the proof for Theorem 5.2 considering a finite, discrete domain $\mathcal{D}$. We start by proving the following inequality that links $f(\boldsymbol{x})$ with the posterior mean and variance of $f$.

**Lemma G.1.** *Pick $\delta \in (0, 1)$ and let $\beta_t = 2\log\left(\frac{|\mathcal{D}|\pi^2 t^2}{6\delta}\right)$. Then, with probability at least $1 - \delta$,*

$$|f(\boldsymbol{x}) - \mu_t(\boldsymbol{x})| \leq \beta_t^{\frac{1}{2}}\left(\sum_{i=1}^n \sqrt{\sum_{k \in \mathcal{N}_i} \frac{\left(\sigma_t^{(k)}(\boldsymbol{x}_{\mathcal{V}_k})\right)^2}{|\mathcal{N}_k|^2}}\right) \tag{57}$$

*for all $\boldsymbol{x} \in \mathcal{D}$ and $t \in \mathbb{N}$. $\mu_t(\boldsymbol{x})$ and $\sigma_t^2(\boldsymbol{x}) = \sum_{i=1}^n \left(\sigma_t^{(i)}(\boldsymbol{x}_{\mathcal{V}_i})\right)^2$ are given by the decomposition in Proposition 3.4.*

*Proof.* For all $\boldsymbol{x} \in \mathcal{D}$ and $t \in \mathbb{N}$, we have $f(\boldsymbol{x}) \sim \mathcal{N}\left(\mu_t(\boldsymbol{x}), \sigma_t^2(\boldsymbol{x})\right)$. Defining $s_t(\boldsymbol{x}) = \frac{f(\boldsymbol{x}) - \mu_t(\boldsymbol{x})}{\sigma_t(\boldsymbol{x})}$, we know that $s_t(\boldsymbol{x}) \sim \mathcal{N}(0,1)$. Therefore we have successively that

$$\Pr\left(|s_t(\boldsymbol{x})| \leq \beta_t^{\frac{1}{2}}\right) \geq 1 - e^{-\frac{\beta_t}{2}}$$

$$\Pr\left(|f(\boldsymbol{x}) - \mu_t(\boldsymbol{x})| \leq \beta_t^{\frac{1}{2}} \sigma_t(\boldsymbol{x})\right) \geq 1 - e^{-\frac{\beta_t}{2}}$$

$$\Pr\left(|f(\boldsymbol{x}) - \mu_t(\boldsymbol{x})| \leq \beta_t^{\frac{1}{2}}\left(\sum_{i=1}^{n} \sqrt{\sum_{k \in \mathcal{N}_i} \frac{\left(\sigma_t^{(k)}(\boldsymbol{x}_{\mathcal{V}_k})\right)^2}{|\mathcal{N}_k|^2}}\right)\right) \geq 1 - e^{-\frac{\beta_t}{2}} \tag{58}$$

where the last inequality (58) follows from (7) (see Theorem 3.5). Inequality (58) holds for one single pair $(t, \boldsymbol{x})$. Applying the union bound for all pairs in $\mathbb{N} \times \mathcal{D}$, we have $\forall t \in \mathbb{N}, \forall \boldsymbol{x} \in \mathcal{D}$

$$\Pr\left(|f(\boldsymbol{x}) - \mu_t(\boldsymbol{x})| \leq \beta_t^{\frac{1}{2}} \sum_{i=1}^{n} \sqrt{\sum_{k \in \mathcal{N}_i} \frac{\left(\sigma_t^{(k)}(\boldsymbol{x}_{\mathcal{V}_k})\right)^2}{|\mathcal{N}_k|^2}}\right) \geq 1 - |\mathcal{D}| \sum_{t=1}^{+\infty} e^{-\frac{\beta_t}{2}}. \tag{59}$$

Pick $\delta \in (0,1)$ and let $\beta_t = 2\log\left(\frac{|\mathcal{D}|\pi^2 t^2}{6\delta}\right)$. Then

$$|\mathcal{D}| \sum_{t=1}^{+\infty} e^{-\frac{\beta_t}{2}} = |\mathcal{D}| \sum_{t=1}^{+\infty} e^{-\log\left(\frac{|\mathcal{D}|\pi^2 t^2}{6\delta}\right)}$$

$$= \frac{6\delta}{\pi^2} \sum_{t=1}^{+\infty} \frac{1}{t^2}$$

$$= \delta.$$

Therefore, (59) becomes

$$\Pr\left(|f(\boldsymbol{x}) - \mu_t(\boldsymbol{x})| \leq \beta_t^{\frac{1}{2}} \sum_{i=1}^{n} \sqrt{\sum_{k \in \mathcal{N}_i} \frac{\left(\sigma_t^{(k)}(\boldsymbol{x}_{\mathcal{V}_k})\right)^2}{|\mathcal{N}_k|^2}}\right) \geq 1 - \delta \tag{60}$$

which is the desired result. $\qquad\square$

We are now ready to bound the immediate regret $r_t = f(\boldsymbol{x}^*) - f(\boldsymbol{x}^t)$ and prove Theorem 5.2.

*Proof.* By definition, $\boldsymbol{x}^t = \arg\max_{\boldsymbol{x} \in \mathcal{D}} \sum_{i=1}^{n} \varphi_t^{(i)}(\boldsymbol{x}_{\mathcal{V}_i})$. Therefore, $\sum_{i=1}^{n} \varphi_t^{(i)}(\boldsymbol{x}_{\mathcal{V}_i}^t) \geq \sum_{i=1}^{n} \varphi_t^{(i)}(\boldsymbol{x}_{\mathcal{V}_i}^*)$ and

$$\sum_{i=1}^{n} \varphi_t^{(i)}(\boldsymbol{x}_{\mathcal{V}_i}^t) = \sum_{i=1}^{n} \mu^{(i)}(\boldsymbol{x}_{\mathcal{V}_i}^t) + \beta_t^{\frac{1}{2}} \sqrt{\sum_{k \in \mathcal{N}_i} \frac{\left(\sigma_t^{(k)}(\boldsymbol{x}_{\mathcal{V}_k}^t)\right)^2}{|\mathcal{N}_k|^2}} \geq \sum_{i=1}^{n} \varphi_t^{(i)}(\boldsymbol{x}_{\mathcal{V}_i'})$$

$$\geq f(\boldsymbol{x}^*) \tag{61}$$

with (61) following from Lemma G.1 with probability $1-\delta$. We can now upper bound the immediate regret $r_t$, since with probability $1-\delta$, we have

$$r_t = f(\boldsymbol{x}^*) - f(\boldsymbol{x}^t)$$

$$\leq \sum_{i=1}^n \mu^{(i)}(\boldsymbol{x}_{\mathcal{V}_i}^t) + \beta_t^{\frac{1}{2}} \sqrt{\sum_{k\in\mathcal{N}_i} \frac{\left(\sigma_t^{(k)}(\boldsymbol{x}_{\mathcal{V}_k}^t)\right)^2}{|\mathcal{N}_k|^2}} - f(\boldsymbol{x}^t)$$

$$= \sum_{i=1}^n \mu^{(i)}(\boldsymbol{x}_{\mathcal{V}_i}^t) - f(\boldsymbol{x}^t) + \beta_t^{\frac{1}{2}} \sqrt{\sum_{k\in\mathcal{N}_i} \frac{\left(\sigma_t^{(k)}(\boldsymbol{x}_{\mathcal{V}_k}^t)\right)^2}{|\mathcal{N}_k|^2}}$$

$$= \mu_t(\boldsymbol{x}^t) - f(\boldsymbol{x}^t) + \beta_t^{\frac{1}{2}} \sum_{i=1}^n \sqrt{\sum_{k\in\mathcal{N}_i} \frac{\left(\sigma_t^{(k)}(\boldsymbol{x}_{\mathcal{V}_k}^t)\right)^2}{|\mathcal{N}_k|^2}}. \tag{62}$$

Combining (57) with (62), we get

$$\Pr\left(r_t \leq 2\beta_t^{\frac{1}{2}} \sum_{i=1}^n \sqrt{\sum_{k\in\mathcal{N}_i} \frac{\left(\sigma_t^{(k)}(\boldsymbol{x}_{\mathcal{V}_k}^t)\right)^2}{|\mathcal{N}_k|^2}}\right) \geq 1-\delta, \tag{63}$$

which is the desired result. $\qquad\square$

## H SYNTHETIC FUNCTIONS

In this section, we describe the synthetic functions constituting our benchmark in Section 6.1.

### H.1 SIX-HUMP CAMEL FUNCTION

The Six-Hump Camel function is a 2-dimensional function defined by

$$f(x_1, x_2) = \left(-4 + 2.1x_1^2 - \frac{x_1^4}{3}\right) x_1^2 - x_1 x_2 + \left(4 - 4x_2^2\right) x_2^2. \tag{64}$$

It is composed of $n = 3$ factors, with a MFS $\bar{d} = 2$. In our experiment, we optimize it on the rectangle $\mathcal{D} = [-3, 3] \times [-2, 2]$. It has 6 local maxima, two of which are global with $f(\boldsymbol{x}^*) = 1.0316$.

### H.2 HARTMANN FUNCTION

The Hartmann function is a 6-dimensional function defined by

$$f(\boldsymbol{x}) = \sum_{i=1}^4 \alpha_i \exp\left(-\sum_{j=1}^6 A_{ij} \left(x_j - P_{ij}\right)^2\right), \tag{65}$$

with $\boldsymbol{\alpha} = (\alpha_i)_{i\in[\![1,4]\!]}$, $\boldsymbol{A} = (A_{ij})_{(i,j)\in[\![1,4]\!]\times[\![1,6]\!]}$ and $\boldsymbol{P} = (P_{ij})_{(i,j)\in[\![1,4]\!]\times[\![1,6]\!]}$ given as constants.

It is composed of $n = 4$ factors, with a MFS $\bar{d} = 6$. In our experiment, we optimize it on the hypercube $\mathcal{D} = [0, 1]^6$. It has 6 local maxima and a global maximum with $f(\boldsymbol{x}^*) = 10.5364$.

### H.3 POWELL FUNCTION

The Powell function is a function of an arbitrary number $d = 4k$ of dimensions, defined by

$$f(\boldsymbol{x}) = -\sum_{i=1}^{d/4} (x_{4i-3} + 10x_{4i-2})^2 + 5(x_{4i-1} - x_{4i})^2 + (x_{4i-2} - 2x_{4i-1})^4 + 10(x_{4i-3} - x_{4i})^4. \tag{66}$$

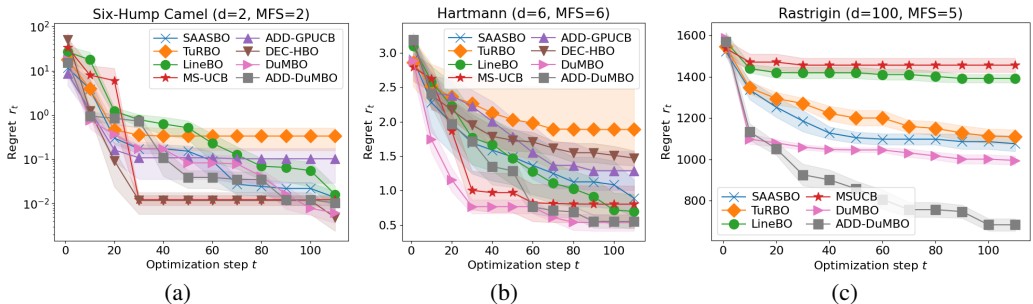

Figure 4: Performance achieved by the studied BO algorithms for (a) the 2d Six-Hump Camel function, (b) the 6d Hartmann function and (c) the 100d Rastrigin function. The shaded areas indicate the standard error intervals.

We chose to set $k = 6$, so that the resulting Powell function lives in a $d = 24$ dimensional space. It is composed of $n = 6$ factors, with a MFS $\bar{d} = 4$. In our experiment, we optimize it on the hypercube $\mathcal{D} = [-4, 5]^{24}$. It has a global maximum at $\boldsymbol{x}^* = \boldsymbol{0}$, with $f(\boldsymbol{x}^*) = 0$.

### H.4 RASTRIGIN FUNCTION

The Rastrigin function is a function of an arbitrary number $d$ of dimensions, defined by

$$f(\boldsymbol{x}) = -10d - \sum_{i=1}^{d} x_i^2 - 10\cos\left(2\pi x_i\right). \tag{67}$$

We chose to set $d = 100$. We also chose to aggregate some factors to make the optimization problem harder. The resulting Rastrigin function is composed of $n = 20$ factors, with a MFS $\bar{d} = 5$. In our experiment, we optimize it on the hypercube $\mathcal{D} = [-5.12, 5.12]^{100}$. It has multiple, regularly distributed local maxima, with a global maximum at $\boldsymbol{x}^* = \boldsymbol{0}$ and $f(\boldsymbol{x}^*) = 0$.

### H.5 ADDITIONAL FIGURES

Figure 4 depicts the performance of the studied BO algorithms on the synthetic functions not discussed in Section 6.1.

Figure 4(a) reports the minimal regrets achieved by the solutions on the Six-Hump Camel (SHC) function. Observe that in this specific example, DuMBO clearly outperforms every other state-of-the-art algorithm except DEC-HBO. This is due to the simplicity of the SHC function that satisfies all the assumptions made by DEC-HBO: a MFS $\bar{d}$ lower than 3 and a sparse factor graph. In this case, the variant of the max-sum algorithm used by DEC-HBO is guaranteed to query $\arg\max \varphi_t$ at each time step $t$.

Figures 4(b) and 4(c) depict dynamics similar to Figure 1(a). In both cases, the ability to infer or exploit a complex additive decomposition gives DuMBO a decisive advantage against the other BO algorithms. As a consequence, DuMBO and ADD-DuMBO manage to outperform them, even in very high dimensional input spaces (see Figure 4(c)). Note that ADD-GPUCB and DEC-HBO were not evaluated on the Rastrigin function, as their execution time exceeded 24 hours because of the large dimensionality of the function.

Finally, observe that LineBO and MS-UCB tend to become less effective as the dimension of the problem grows. Indeed, they adopt strategies (random line searches and discrete sampling in high-dimensional spaces) that are very sensitive to the dimension of the problem.

### I REAL-WORLD PROBLEMS

In this appendix, we describe the real-world problems constituting our benchmark in Section 6.2.

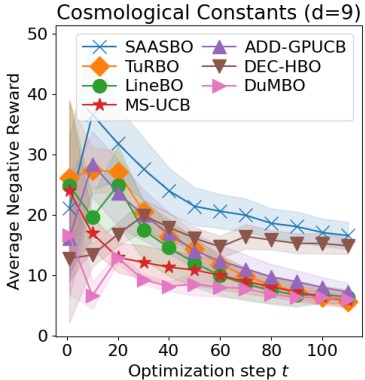

Figure 5: Performance of the studied BO algorithms on the cosmological constants fine-tuning problem.

## I.1 COSMOLOGICAL CONSTANTS

The cosmological constants problem consists in fine-tuning an astrophysics tool to optimize the likelihood of some observed data. We chose to compute the likelihood of the galaxy clustering in Chuang et al. (2013) from the Data Release 9 (DR9) CMASS sample of the SDSS-III Baryon Oscillation Spectroscopic Survey (BOSS). To compute the likelihood, we instrumented the cosmological parameter estimation code CosmoSIS (Zuntz et al., 2015)[2].

We used nine cosmological constants in our optimization task, going from the Hubble's constant to the mass of the neutrinos. If a BO algorithm provided a set of inconsistent cosmological constants, a likelihood of $y = -60$ was returned.

Note that similar experiments were described in other works, such as Kandasamy et al. (2015); Eriksson et al. (2019). However, they were conducted on another, older dataset, with a deprecated NASA simulator[3]. This makes the conducted experiments painful to reproduce on a modern computer. Luckily, CosmoSIS is well documented and easier to install and instrument, so we conducted our experiment with CosmoSIS to make it easier to replicate.

Figure 5 depicts the performance of the described BO algorithms on this problem. Note that, since the objective function does not have an additive decomposition, ADD-DuMBO cannot be evaluated. Although the objective function does not have an additive decomposition, DuMBO demonstrates its competitiveness by achieving the best performance, along with TuRBO, LineBO and MS-UCB.

## I.2 SHANNON CAPACITY OF A WLAN

The Shannon capacity (Kemperman, 1974) sets a theoretical upper bound on the throughput of a wireless communication, depending on the Signal-to-Interference plus Noise Ratio (SINR) of the communication. Denoting by $S_{i,j}$ the SINR between two wireless devices $i$ and $j$ communicating on a radio channel of bandwidth $W$ (in Hz), the Shannon capacity $C(S_{i,j})$ (in bits) is defined by

$$C(S_{i,j}) = W \log_2 (1 + S_{i,j}).$$ (68)

In this problem, we study a Wireless Local Area Network (WLAN) with end-users associated to nodes streaming a continuous, large amount of data. The WLAN topology is depicted in Figure 6. It is populated with 36 end-users, each one associated to one of the 12 depicted nodes. Note that each node is within the radio range of at least two other nodes. This creates interference and, consequently, reduces the SINRs between nodes and end-users.

Each node has an adjustable transmission power $x_i \in [10^{0.1}, 10^{2.5}]$ in mW (milliwatts). This task consists in jointly optimizing the Shannon capacity (68) of each pair (node, associated end-user) by

---

[2]https://cosmosis.readthedocs.io/en/latest/reference/standard_library/BOSS.html
[3]https://lambda.gsfc.nasa.gov/toolbox/lrgdr/

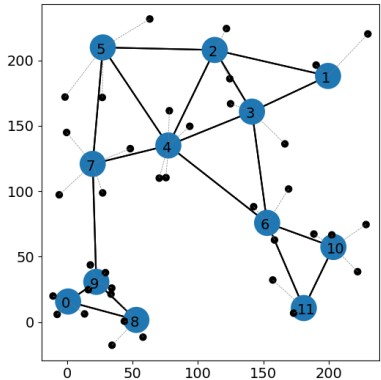

Figure 6: The WLAN topology used in the Shannon capacity optimization experiment. The end-users are depicted as black dots, the nodes as numbered blue circles and the associations between end-users and nodes as thin gray lines. Two nodes are connected with a black line if they are within the radio range of each other.

tuning the transmission power of the nodes. That is, the objective function $f$ is a 12-dimensional function defined by

$$f(\boldsymbol{x}) = \sum_{i=1}^{12} \sum_{j \in \mathcal{N}_i} C(S_{i,j}), \tag{69}$$

with $\mathcal{N}_i$ the set of end-users associated to node $i$.

A difficult trade-off needs to be found because a node cannot simply use the maximum transmission power as this would cause a lot of interference for the neighboring nodes. Given a configuration $\boldsymbol{x} \in \mathcal{D} = [10^{0.1}, 10^{2.5}]^{12}$, the SINRs are provided by the well-recognized network simulator ns-3 (The ns3 Project) that reliably reproduces the WLAN internal dynamics. The additive decomposition comprises $n = 12$ factors, with a MFS of $\bar{d} = 5$, obtained by making the reasonable assumption that only the neighboring nodes of node $i$ (*i.e.* those within the radio range of node $i$) are creating interference for the communications of node $i$.

### I.3 ROVER TRAJECTORY PLANNING

This problem was also considered by Eriksson et al. (2019); Wang et al. (2018). The goal is to optimize the trajectory of a rover from a starting point $\boldsymbol{s} \in [0, 1]^2$ to a target $\boldsymbol{t} \in [0, 1]^2$, over a rough terrain.

The trajectory is defined by a vector of $d = 60$ dimensions, reshaped into 30 2-d points in $[0, 1]$. A B-spline is fitted to these 30 points, determining the trajectory of the rover. The objective function to optimize is

$$f(\boldsymbol{x}) = -c(\boldsymbol{x}) - 10(||\boldsymbol{x}_{0,1} - \boldsymbol{s}||_1 + ||\boldsymbol{x}_{59,60} - \boldsymbol{t}||_1), \tag{70}$$

with $c(\boldsymbol{x})$ the cost of the trajectory, obtained by integrating the terrain roughness function over the B-spline, and the two $L_1$-norms serving as incentives to start the trajectory near $\boldsymbol{s}$, and to end it near $\boldsymbol{t}$.

## J   WALL-CLOCK TIME

In this section, we provide wall-clock time measurements (excluding the evaluation time of the objective function) of the described BO algorithms on a synthetic function (24d Powell) and a real-world problem (WLAN) described in Appendices H and I respectively. The measurements were taken using a server equipped with two Intel(R) Xeon(R) CPU E5-2690 v4 @ 2.60GHz, with 14 cores (28 threads) each.

Figure 7 gathers the wall-clock time measurements. Observe that DuMBO does not only offer very competitive performance, it also exhibits a lower overhead when compared to the other decompos-

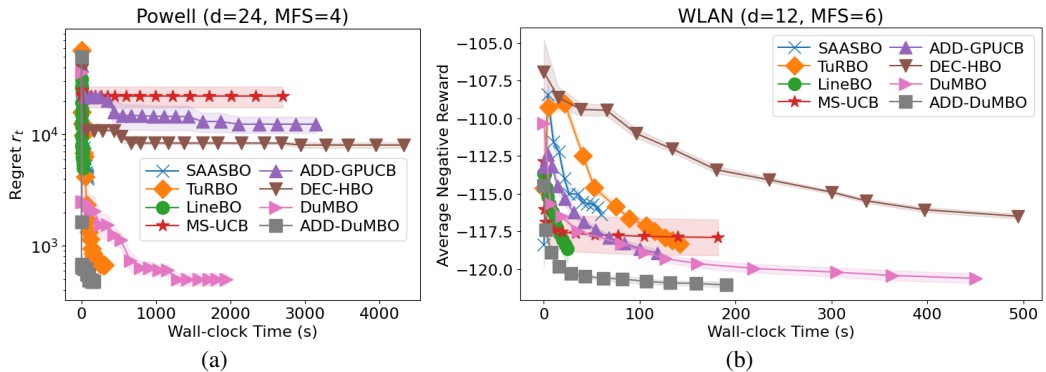

Figure 7: Performance achieved by all the described BO algorithms (including the two versions of DuMBO) for (a) the 24d Powell synthetic function and (b) the maximization of the Shannon capacity in a WLAN. The shaded areas indicate the standard error intervals.

ing algorithms (DEC-HBO and ADD-GPUCB). However, SAASBO, TuRBO and LineBO manage to get lower runtimes than DuMBO. This is not surprising since, by design, these methods have minimal overheads, at the expense of weakening their theoretical guarantees. As for MS-UCB, it manages to have a lower execution time on the WLAN problem (where $d = 12$), but a larger execution time on Powell (where $d = 24$).

Nevertheless, with ADD-DuMBO, having access to the true additive decomposition of the function reduces the overhead of the solution, because the decomposition does not need to be inferred anymore. On the Powell synthetic function (see Figure 7(a)), ADD-DuMBO even achieves the lowest wall-clock time.

