# OpenReview forum: "Relaxing the Additivity Constraints in Decentralized No-Regret High-Dimensional Bayesian Optimization"
_ICLR.cc/2024/Conference — ICLR 2024 poster_

### Official Review · Reviewer_PPCk · 2023-10-29

**Soundness:** 3 good
**Presentation:** 2 fair
**Contribution:** 2 fair
**Rating:** 5
**Confidence:** 4

**Summary:**

The work discusses a new algorithm called DumBO for decentralized Bayesian optimization. DumBO relaxes the additivity constraints of traditional BO algorithms for high-dimensional inputs. It achieves better performance against some existing BO algorithms, especially when the additive structure of the objective function comprises high-dimensional factors. DumBO finds the maximum of the objective function while ensuring consistency between shared input components. The algorithm relies on ADMM to maximize the acquisition function and has demonstrated good performance in the numerical experiments.

**Strengths:**

The authors have conducted a number of numerical experiments to illustrate the practicability of their algorithm. They have included state-of-the-art baseline algorithms for comparison, and have demonstrated scenarios when their approach is superior.

In addition, the authors provide explicit descriptions of optimizing the acquisition function, which is facilitated by the ADMM algorithm. The detailed procedure makes the algorithm feasible to use.

**Weaknesses:**

The presentation of the manuscript is not explicit enough for the audience to understand. For example, the main assumption of this work is that the objective function $f$ can be decomposed into several factors and each factor only takes some dimensions of $\boldsymbol{x}$ as the input, as in Equation (2). However, this decomposition is not explicitly explained in Section 3.1. Actually, Figure 2 in the supplements helps the audience to understand and can be moved to the main text. In addition, to guarantee the performance of ADMM, the authors assume that the acquisition function ``is a restricted prox-regular function that satisfies the Kurdyka-Lojasiewicz condition’’. The authors should at least provide a formal definition of this condition.

The theoretical results do not exhibit sufficient novelty. For example, similar results as Theorem 3.4 have been derived, for example, in [1]. The analysis of the regret is also quite similar to that in [2], without significant novelty.

References:
[1] Mutny, Mojmir, and Andreas Krause. "No-regret algorithms for capturing events in Poisson point processes." In International Conference on Machine Learning, pp. 7894-7904. PMLR, 2021.
[2] Srinivas, Niranjan, Andreas Krause, Sham Kakade, and Matthias Seeger. "Gaussian process optimization in the bandit setting: no regret and experimental design." In Proceedings of the 27th International Conference on International Conference on Machine Learning, pp. 1015-1022. 2010.
[3] Kandasamy, Kirthevasan, Jeff Schneider, and Barnabás Póczos. "High dimensional Bayesian optimisation and bandits via additive models." In International conference on machine learning, pp. 295-304. PMLR, 2015.

**Questions:**

In addition, the authors mention that the decomposition needs to be inferred from the data. In this way, is the algorithm performance sensitive to the inference results of the decomposition? How to incorporate the inference uncertainty into the regret analysis? If the data for inference is not given in advance, how many rounds are required to generate the data for decomposition inference?

Besides, the acquisition functions should satisfy a certain condition as in Assumption 5.1. The authors might discuss under what conditions the acquisition function satisfies Assumption 5.1.

Besides, the authors might provide a more detailed comparison with the existing algorithm Add-GP-UCB [3], since the decomposition of the objective function is quite similar.

---

> ### Author Response · Authors · 2023-11-14
>
> We thank you for the detailed review. We address the comments and questions you have raised below:
>
> **This decomposition is not explicitly explained in Section 3.1**
>
> Thank you for the suggestion. Figure 2 was not included in the main text for lack of space. Under acceptance, we will try to make room for it or to improve the presentation of the additive decomposition.
>
> **The authors assume that the acquisition function is a restricted prox-regular function that satisfies the KL condition’’. The authors should provide a formal definition of this condition. [...] The authors might discuss under what conditions the acquisition function satisfies this assumption.**
>
> In the updated PDF, we prove that this assumption (Assumption 5.1 in the previous version of the paper) always holds under a very mild assumption on the Hessian matrix of the kernel (Assumption 3.1 in the updated PDF), which is met by the majority of common kernel functions used in BO. Assumption 5.1 has now become Theorem 5.1 in the updated PDF, and is proven in Appendix F.
>
> **The theoretical results do not exhibit sufficient novelty.**
>
> As mentioned above, please refer to the updated PDF as we believe our new theoretical result (Theorem 5.1) significantly strengthens the theoretical value of the paper. It removes the potentially strong assumption on the acquisition function (Assumption 5.1 in the previous version of the paper), replacing it by a much weaker assumption (Assumption 3.3 in the updated PDF) that is met by the majority of common kernel functions used in BO.
>
> **Is the algorithm performance sensitive to the inference results of the decomposition? How to incorporate the inference uncertainty into the regret analysis?**
>
> Indeed, the quality of the decomposition will impact the performance of the algorithm. When the additive decomposition is unknown, [1] is used to infer it. Under the hood, [1] uses a Monte-Carlo Markov Chain (MCMC) method, and more specifically a Metropolis-Hastings algorithm. As far as we know, it does not provide any uncertainty criterion.
>
> Using the absolute difference of log-likelihoods between decompositions as a distance on the space of possible additive decompositions could perhaps allow us to define an uncertainty criterion on the decomposition. We did not explore this in the current work, but it is an interesting extension to our paper, and we will definitely consider it for future work. Thank you for the suggestion.
>
> **How many rounds are required to generate the data for decomposition inference?**
>
> The experiments in [1] suggest that, under a hundred iterations, the MCMC approach is able to match the performance obtained from the true additive decomposition.
>
> **The authors might provide a more detailed comparison with the existing algorithm Add-GP-UCB, since the decomposition of the objective function is quite similar.**
>
> In our opinion, ADD-GP-UCB and DuMBO are not really similar. Unlike DuMBO, ADD-GP-UCB assumes a very low $\bar d$ with orthogonal domains. Note that a similar comment can be made for DEC-HBO (although the MFS does not have to be as low as the one required by ADD-GP-UCB, it still needs to be quite low, which tends to limit the complexity of the additive decomposition inference).
>
> Conversely, our algorithm does not make any assumption on the additive structure of $f$. This feature is actually one of the main strengths of our algorithm, as it allows DuMBO to infer arbitrarily complex additive decomposition, and to outperform significantly both ADD-GP-UCB and DEC-HBO (please refer to Table 2 or, more specifically, to Figures 1(a-c) and 4(b)).
>
> [1] Gardner, J., Guo, C., Weinberger, K., Garnett, R., & Grosse, R. (2017, April). Discovering and exploiting additive structure for Bayesian optimization. In Artificial Intelligence and Statistics (pp. 1311-1319). PMLR.

---

> ### Author Response · Authors · 2023-11-20
>
> Dear reviewer,
>
> This is a gentle reminder that our rebuttal as well as our updated paper are online.
>
> In particular, we draw your attention on Theorem 5.1 that addresses a concern you have raised and that, in our opinion, significantly strengthens the theoretical contribution of this paper by establishing the asymptotical optimality of DuMBO under a very mild assumption (Assumption 3.3). We also added more baselines to our numerical results (see Section 6) to strengthen our experiments.
>
> We understand that your time is very valuable, but as the discussion period ends in a few days, we kindly ask you to consider providing any feedback and/or increasing your score. We are of course ready to discuss any remaining concern you may have.

---

### Official Review · Reviewer_NC5W · 2023-10-30

**Soundness:** 3 good
**Presentation:** 3 good
**Contribution:** 2 fair
**Rating:** 8
**Confidence:** 3

**Summary:**

The paper is concerned with the problem of optimizing a function that is noisy and costly to evaluate. The traditional way to tackle this is by assuming an additive structure for the function which imposes restrictive assumptions on the function. This paper introduces DumBO, a decentralized BO algorithm that relaxes these assumptions, albeit at the expense of weakening the maximization guarantees of the acquisition function. Additionally, the authors also claim to address the over-exploration problem in decentralized BO algorithms. Experimental evaluation suggest that their algorithm performs competitively compared to state-of-the-art BO algorithms.

**Strengths:**

1) The paper is very well-written.
2) The authors provide an algorithm which can be implemented in a decentralized manner, which seems to be very useful.
3) The authors show a regret bound, which  show asymptotic optimality of their algorithm.
4) Experiments conducted are exhaustive.

**Weaknesses:**

1) It would have been nice if the authors could show a proof sketch of their main theoretical result.
2) Apart from the proof, it would be nice if the authors could highlight the main contributions and the technical (or analytical) challenges they faced.
These all are not major weaknesses per se, but can help the interested readers.

**Questions:**

1) Are there known lower bounds for the problem? If yes, it would be nice to see how the proposed algorithm does in comparison.
2) What were the major technical challenges the authors faced while deriving the regret bounds?

---

> ### Author Response · Authors · 2023-11-14
>
> We thank you for the detailed review. We address the comments and questions you have raised below:
>
> **It would have been nice if the authors could show a proof sketch of their main theoretical result.**
>
> Thank you for the suggestion. A proof sketch was not included in the main text because of lack of space, but we will include it in the camera-ready version, space permitting.
>
> **It would be nice if the authors could highlight the main contributions and the technical (or analytical) challenges they faced. [...] What were the major technical challenges the authors faced while deriving the regret bounds?**
>
> Thank you for the suggestion. Our main analytical challenge was embodied by Assumption 5.1, but has been lifted now as it became Theorem 5.1. We kindly refer you to the updated PDF for this new result that completes the theoretical proof of the asymptotic optimality of DuMBO. Under acceptance, we will try to make room for highlighting the contributions and the challenges we faced.
>
> **Are there known lower bounds for the problem? If yes, it would be nice to see how the proposed algorithm does in comparison.**
>
> We are not aware of a lower bound on the regret for this problem, especially under the additive assumption. If you have any references to this matter, we will be happy to further investigate this point and discuss it with you.

---

> > ### Comment · Reviewer_NC5W · 2023-11-17
> > **Response to authors' rebuttal.**
> >
> > I thank the authors for providing the response. I will keep the rating as is.

---

### Official Review · Reviewer_8R9C · 2023-11-01

**Soundness:** 3 good
**Presentation:** 3 good
**Contribution:** 2 fair
**Rating:** 5
**Confidence:** 4

**Summary:**

The paper relaxes restrictive assumptions on the additive structure of the objective function. It proposes the DuMBO algorithm, which is a decentralized, message-passing, and asymptotically optimal BO algorithm. DuMBO can infer complex additive decompositions of the objective function without assumptions regarding Maximum Factor Size (MFS).

**Strengths:**

The paper presents an innovative approach to Bayesian Optimization (BO) that relaxes assumptions about the Maximum Factor Size (MFS) and focuses on modeling and optimizing complex, high-dimensional objective functions.
The DuMBO algorithm introduced in the paper is asymptotically optimal, meaning it provides strong guarantees of convergence to the global optimum of the objective function over time.
The paper backs its claims with empirical evidence by comparing DuMBO with state-of-the-art BO algorithms on both synthetic and real-world problems. It demonstrates that DuMBO performs competitively and is particularly effective when the objective function comprises numerous factors with a large MFS.
The paper explores the trade-off between model complexity and the guarantee of maximization in the acquisition function, contributing to the theoretical understanding of BO algorithms in high-dimensional spaces.

**Weaknesses:**

The paper's experiment part is not valid enough. The baseline it compares TurBo is not that up to date. Please see:

Learning Search Space Partition for Black-box Optimization using Monte Carlo Tree Search
Linnan Wang, Rodrigo Fonseca, Yuandong Tian

And some other trust region based high-dimensional methods. Also notice that in the paper, some other methods are mentioned, but didn't show up in the experiments part.

In general, the performance improvement to other baseline is not significant enough. I believe some other methods can outperform the baselines in a similar way.

Ablation study about other hyper parameters should be given.

**Questions:**

(1) In Table 2, why the standard deviation is not given?
(2) Is the algorithm sensitive to \bar d and other hyper-parameters?
(3) In figure 1c. The performance of DumBO is close to TurBO. Does that mean DumBO does not outperform baseline much on higher dimensional tasks?
(4) Why the time step is only about 100? For figure 1a. Some methods's cost are still decaying.

---

> ### Author Response · Authors · 2023-11-14
>
> We thank you for the detailed review. We address the comments and questions you have raised below:
>
> **The paper's experiment part is not valid enough. The baseline it compares TurBo is not that up to date. Also notice that in the paper, some other methods are mentioned, but didn't show up in the experiments part.**
>
> Thank you for bringing up this additional reference. Note that this paper proposes to use a meta-algorithm (as coined by the authors themselves) that relies on other high-dimensional BO algorithms (and especially TuRBO) for their sampling needs. In contrast, our paper proposes a no-regret BO algorithm, DuMBO, that solves high-dimensional problems by assuming an additive decomposition of the objective function. This is why we compare DuMBO directly to other no-regret BO algorithms, including TuRBO.
>
> To fully address your concerns, we added two other baselines in our experiments. Please refer to Figures 1, 4 and 5 in the updated PDF.
>
> **In general, the performance improvement to other baseline is not significant enough.**
>
> With all due respect, we beg to differ. The results in the updated PDF show that DuMBO consistently outperforms all the 6 other baselines by more than a confidence interval in 4 out of 7 benchmarks. In the 3 remaining benchmarks, DuMBO systematically belongs to the set of top performing BO algorithms. This observation is further illustrated by Figure 1 in the main text and Figures 4 and 5 in the appendix.
>
> Moreover, when the additive structure of the function is known, DuMBO can exploit this additional information, and obtain performance that are even better (see Table 2, line ADD-DuMBO).
>
> **Ablation study about other hyperparameters should be given.**
>
> The only hyperparameter of our method is the parameter $\eta$ in the augmented Lagrangian (see (11), or (12) in the updated PDF). Indeed $\bar d$ is computed by the algorithm (see below), and the kernel that is used is a well-known Matérn kernel, which is not a hyper-parameter specific to DuMBO.
>
> Note that DuMBO runs without having to adjust hyperparameters by hand. Indeed $\eta$ itself does not actually have to be set as a hyper-parameter, because it can be adjusted online, during the course of ADMM, without hindering the convergence guarantees. The detailed procedure is specified in [1], and this is indeed what we do in practice. We added a sentence regarding the value of $\eta$ under Equation (12) in the updated PDF. Thank you for pointing it out.
>
> **In Table 2, why the standard deviation is not given?**
>
> The standard deviations were not provided in Table 2 only to maintain its readability, because of lack of space. But they are displayed in all the relevant figures (please refer to Figures 1, 4, 5 and 7). Under acceptance, we could try to make room for them. We would be happy to discuss ways to include them in the main text, if you have some suggestions.
>
> **Is the algorithm sensitive to $\bar d$ and other hyper-parameters?**
>
> As $\bar d$ grows, the underlying additive decomposition of the objective function $f$ becomes more complex, in the sense that at least one factor increases its dimensionality. However, since we relaxed any assumption on $\bar d$ (and we do not require the user to provide it), Theorems 5.1 and 5.2 as well as Corollary 5.3 state that $\bar d$ will not influence the convergence in a significant manner. In fact, the larger $\bar d$ is, the better DuMBO performs when compared to the other additive no-regret BO algorithms. Conversely, when $\bar d$ is low, the assumptions made by other additive no-regret BO algorithms become more valid (or at least, less restrictive) so that their performance and DuMBO’s performance tend to become similar. This is pointed out in the comment of Figure 4(a), in the supplementary material.
>
> **In figure 1c. The performance of DuMBO is close to TurBO. Does that mean DuMBO does not outperform baseline much on higher dimensional tasks?**
>
> Unfortunately, we have to move the experiments discussing this point in the supplementary material because of lack of space. Among them, we compared the baselines on the Rastrigin function, with $d=100$ (Figure 4(c)): DuMBO significantly outperforms TuRBO in this example. Note that, unlike TuRBO, DuMBO is proven to be asymptotically optimal. Hence, in the long run, DuMBO’s asymptotic performance will be at least as good as those from TuRBO’s. In the short run, our experiments show that the transient performance of DuMBO is better in 5 out of the 7 benchmarks (see Table 2 and Figures 1 and 4).

---

> ### Author Response · Authors · 2023-11-14
>
> **Why the time step is only about 100? For figure 1a. Some methods's cost are still decaying.**
>
> As mentioned at the beginning of Section 6, BO is usually applied for the optimization of expensive black-box functions. Hence we put the focus on the performance of all algorithms on the first hundred iterations. Note that this is a choice made by some previous works (e.g. [2, 3]). The rationale for this choice is that it is precisely in this first transient phase that methods that have not been shown asymptotically optimal can outperform their asymptotically optimal counterparts (since for the latter the theory predicts, by definition, a global convergence with a sublinear regret). Furthermore, rapid convergence matters for real-time applications. Nevertheless, to fully address your concerns, we have extended the number of iterations for Figure 1(a). The performance of all the baselines with this new horizon can be found in Table 2 (Powell column) and in Figure 1(a) in the updated PDF.
>
> [1] Boyd, S., Parikh, N., Chu, E., Peleato, B., & Eckstein, J. (2011). Distributed optimization and statistical learning via the alternating direction method of multipliers. Foundations and Trends in Machine learning, 3(1), 1-122.
>
> [2] Gupta, S., Rana, S., & Venkatesh, S. (2020, April). Trading convergence rate with computational budget in high dimensional Bayesian optimization. In Proceedings of the AAAI Conference on Artificial Intelligence (Vol. 34, No. 03, pp. 2425-2432).
>
> [3] Hoang, T. N., Hoang, Q. M., Ouyang, R., & Low, K. H. (2018, April). Decentralized high-dimensional Bayesian optimization with factor graphs. In Proceedings of the AAAI Conference on Artificial Intelligence (Vol. 32, No. 1).

---

### Official Review · Reviewer_q9mg · 2023-11-03

**Soundness:** 2 fair
**Presentation:** 2 fair
**Contribution:** 2 fair
**Rating:** 5
**Confidence:** 3

**Summary:**

This paper addresses a line of the high-dimensional Bayesian optimization problem under the assumption that the objective function has an additive structure. Under this line, the authors relax the assumption of the additive structure without requiring that the maximum number of dimensions for a factor of the decomposition is low as in the work of (Hoang et al., 2018). The authors propose DuMBO, a decentralized, message-passing, provably asymptotically optimal Bayesian optimization algorithm under such a relaxed assumption. They also introduce another way to approximate the GP-UCB acquisition function. Finally, they demonstrate the effectiveness of DuMBO by comparing it with several state-of-the-art BO algorithms on both synthetic and real-world problems.

**Strengths:**

- The paper is well written in general although the related works are missing and the introduction section can be improved.

- The idea of using the Alternating Direction Method of Multipliers (ADMM) proposed by Gabay & Mercier (1976) to maximize the acquisition function is new compared to the existing works.

**Weaknesses:**

- The related works are missing. Besides Embedding and Decomposing and Turbo, there are several approaches as in [1], [2], and [3] that do not impose assumptions on the structure of the function $f$. In particular, the authors are missing a very related paper [4] "Are Random Decompositions All We Need in High-Dimensional Bayesian Optimisation?".  A comparison with this work is needed.

- The representation in the method is unclear to understand the contribution of this paper. For example, it is unclear to me why ADMM can solve the high-dimensional optimization problem, for example when $\overline{d}$ is high. Could we ensure to find the arg max when the function is non-convex?

- The experiments are insufficient to understand the effectiveness of the proposed method.  It is missing a baseline from [4].

[1]. Johannes Kirschner,Mojm´ır Mutny', Nicole Hiller, Rasmus Ischebeck, Andreas Krause. Adaptive and Safe Bayesian Optimization in High Dimensions via One-Dimensional Subspaces. ICML 2019.

[2]. Hung Tran-The, Sunil Gupta, Santu Rana, Svetha Venkatesh. Trading Convergence Rate with Computational Budget in High Dimensional Bayesian Optimization. AAAI 2020.

[3]. Linnan Wang, Rodrigo Fonseca, Yuandong Tian. Learning Search Space Partition for Black-box
Optimization using Monte Carlo Tree Search. NeurIPS 2021.

[4]. Juliusz Ziomek, Haitham Bou-Ammar. Are Random Decompositions all we need in High Dimensional Bayesian
Optimisation? ICML 2023.

**Questions:**

Please see my above questions. I also have other questions as follows:

- Do the authors need any assumption on the function $f$ such as the Lipschitz continuous to guarantee the convergence?
- Equation (8) seems incorrect to me. Is it a typo?

---

> ### Author Response · Authors · 2023-11-14
>
> We thank you for the detailed review. We address the comments and questions you have raised below:
>
> **The related works are missing [...] that do not impose assumptions on the structure of the function.**
>
> Thank you for bringing up these additional references. We included them in our Related Works section, and we also discuss them below.
>
> [1] This article introduces LineBO. Although the core idea in this article (successive line searches with random directions) does not impose any assumption on the structure of the function and allows LineBO to find its next query very rapidly, the algorithm does not provide a no-regret guarantee. We implemented LineBO and added it to our baselines. After testing this algorithm with our benchmarks, it seems that searching along a line becomes less effective as the dimension of the domain grows. Please refer to Table 2 and Figures 1, 4 and 5 in the updated PDF for the additional experimental results. Note that, unlike LineBO, DuMBO provides a no-regret guarantee and is less sensitive to the dimension of the problem.
>
> [2] This paper proposes MS-UCB. Its core idea is to perform the optimization of the acquisition function within two orthogonal, lower-dimensional subspaces $\mathcal{Y}$ and $\mathcal{Z}$ of dimension $d$ and $D - d$, respectively, instead of the original domain $\mathcal{X}$ of dimension $D$. MS-UCB is asymptotically optimal and does not impose any assumption on the structure of $f$. However, for the acquisition maximization, MS-UCB still uses a computationally expensive algorithm on subspace $\mathcal{Y}$ (this imposes $d << D$ in practice) and requires $O(t^{\alpha+1})$ GP inferences at each iteration $t$ on subspace $\mathcal{Z}$. If $\alpha$ is low, the cumulative regret is still sublinear but the average regret decreases quite slowly (see Theorem 2 in [2]). If $\alpha$ is large, MS-UCB becomes very slow in practice. We implemented MS-UCB and added it to our baselines. After testing this algorithm with our benchmarks, it appears that, similarly to LineBO, MS-UCB is very sensitive to the dimension of the problem. In fact, the acquisition function cannot be quickly maximized in $\mathcal{Z}$ without increasing $\alpha$ and drastically increasing the execution time. Please refer to Table 2 and Figures 1, 4 and 5 in the updated PDF for the additional experimental results. In contrast, DuMBO offers the same no-regret guarantee, but is decentralized, has a lower computational complexity, fewer hyperparameters and is able to exploit additional information about the problem (i.e., the additive structure of $f$, if it exists).
>
> [3] This paper proposes to use a high-dimensional Bayesian optimization algorithm (especially TuRBO) for their sampling needs: the authors do not propose a BO algorithm per se but a meta-algorithm (as coined by the authors) that relies on other BO algorithms. In contrast, our paper proposes a no-regret BO algorithm, DuMBO, that solves high-dimensional problems by assuming an additive decomposition of the objective function. This is why we compare DuMBO directly to other BO algorithms, including TuRBO.
>
> [4] Thank you for pointing us to this IMCL 2023 paper that appeared at the end of July. We are happy to add it in the related work section. Note that [4] appeared less than 2 months before the submission deadline and is considered by ICLR as contemporaneous. ICLR reviewing policy (see last item in the FAQ at https://iclr.cc/Conferences/2024/ReviewerGuide states indeed “if a paper was published (i.e., at a peer-reviewed venue) on or after May 28, 2023, authors are not required to compare their own work to that paper”.
>
> Nonetheless, we are also happy to engage in the discussion on that paper. It is based on the observation that decompositions that are learnt from data can be suboptimal. The authors conduct a regret analysis showing that, in an adversarial context, using random decompositions is on average the best way to lower the cumulative regret of a decomposing BO algorithm. However, they do not propose an algorithm that maximizes an acquisition function in an arbitrary dimensional space with a reasonable computational complexity. The paper is actually quite complementary to our work. In fact, instead of using [5] to infer an additive decomposition, the idea of random decompositions could be used in conjunction with our results (Theorems 3.5, 5.1, 5.2 and Corollary 5.3 in the updated PDF). This would increase the performance of the resulting BO algorithm in cases where the decomposition is hard to infer (although it would degrade its performance in other contexts). This is a very interesting extension and we will definitely consider it for future work. Thank you again for mentioning this reference.
>
> [5] Gardner, J., Guo, C., Weinberger, K., Garnett, R., & Grosse, R. (2017, April). Discovering and exploiting additive structure for Bayesian optimization. In Artificial Intelligence and Statistics (pp. 1311-1319). PMLR.

---

> ### Author Response · Authors · 2023-11-14
>
> **Could we ensure to find the arg max when the function is non-convex? Do the authors need any assumption on the function $f$ such as the Lipschitz continuous to guarantee the convergence?**
>
> We do not need the function $f$ to be Lipschitz continuous because the assumptions given in Section 3.1, namely Assumption 3.1 and Assumption 3.3 in the updated PDF, provide sufficient regularity.
>
> **The experiments are insufficient to understand the effectiveness of the proposed method.**
>
> To fully address this concern, we implemented LineBO and MS-UCB and we added them to our baselines. The results (see Table 2) show that DuMBO consistently outperforms all the 6 baselines by a margin larger than one confidence interval on 4 out of 7 benchmarks. In the remaining 3 benchmarks, DuMBO is consistently in the set of the top performing BO algorithms. Moreover, when the additive structure of the function is known, DuMBO can exploit this additional information and obtain performance that is even better (see Table 2, line ADD-DuMBO).
>
> **Equation (8) seems incorrect to me. Is it a typo?**
>
> Thank you for pointing out (8). It is indeed a typo, that we corrected in the updated PDF.

---

> ### Author Response · Authors · 2023-11-20
>
> Dear reviewer,
>
> This is a gentle reminder that our rebuttal as well as our updated paper are online.
>
> In particular, we draw your attention on Theorem 5.1 that addresses a concern you have raised and that, in our opinion, significantly strengthens the theoretical contribution of this paper by establishing the asymptotical optimality of DuMBO under a very mild assumption (Assumption 3.3). We also added more baselines to our numerical results (see Section 6) to strengthen our experiments.
>
> We understand that your time is very valuable, but as the discussion period ends in a few days, we kindly ask you to consider providing any feedback and/or increasing your score. We are of course ready to discuss any remaining concern you may have.

---

### Author Response · Authors · 2023-11-14

We thank all the reviewers for their detailed reviews.

Some among you (in particular reviewers q9mg and PPCk) expressed concerns regarding the theoretical results and Assumption 5.1. In the paper, we had indeed proven the asymptotic optimality of DuMBO, but only if Assumption 5.1 was true. During the reviewing period, we managed to prove that Assumption 5.1 actually always holds true under a very mild assumption on the Hessian matrix of the kernel, which is met by a majority of common kernel functions used in BO.

The updated PDF includes these new strengthened results. Assumption 5.1 has now become Theorem 5.1 in the updated PDF, holding under Assumption 3.3. Its proof is provided in Appendix F. We believe that this result significantly strengthens the paper quality and potential impact, since the asymptotic optimality of the algorithm is now proven under mild assumptions that are often met in practice. We hope that this addresses your concerns on the theory front.

---

### Meta-Review · Area_Chair_GDxH · 2023-12-07

**Metareview:**

The authors study the important setting of high-dimensional Bayesian optimization. In order to sidestep the curse of dimensionality in this setting, it is common to assume some additive structure on the objective function into components only depending on low-dimensional subspaces of the domain. The authors describe a decentralized approach for optimization in this setting that is somewhat more flexible than existing approaches. The authors also provide a regret bound for their proposed algorithm and evaluate its performance in a series of experiments with a variety of structure in the objective function (or lacking therefrom).

The reviewers were somewhat split in their support of this paper. They shared a consensus regarding the importance of this problem and enthusiasm for the general approach proposed by the authors.

The main weaknesses noted by some reviewers were:

- a lack of discussion of related methods
- some concerns regarding assumptions required in the theoretical analysis

I believe both issues were addressed satisfactorily by the authors, who clarified the connection to the existing work identified by the reviewers and also provided convincing theoretical support for their assumption 5.1.

**Justification For Why Not Higher Score:**

Although I believe the outcomes of the author-reviewer discussion pushed this paper over the bar for acceptance, the enthusiasm from the reviewers was still relatively muted. Additionally, no explicit champion for the work emerged during the reviewer discussion period.

**Justification For Why Not Lower Score:**

In my opinion, the author-reviewer discussion strengthened the paper significantly, especially the theoretical contributions that were initially in question.

---

### Decision · Program_Chairs · 2024-01-16

Accept (poster)